# Defining the impact of dietary macronutrient balance on PCOS traits

Valentina Rodriguez Paris [1], Samantha M. Solon-Biet [2], Alistair M. Senior [2], Melissa C. Edwards[1,3], Reena Desai[3], Nicodemus Tedla[4], Madeleine J. Cox [1], William L. Ledger [1], Robert B. Gilchrist[1], Stephen J. Simpson [2], David J. Handelsman [3] & Kirsty A. Walters [1,3]✉

Lifestyle, mainly dietary, interventions are first-line treatment for women with polycystic ovary syndrome (PCOS), but the optimal diet remains undefined. We combined a hyperandrogenized PCOS mouse model with a systematic macronutrient approach, to elucidate the impact of dietary macronutrients on the development of PCOS. We identify that an optimum dietary macronutrient balance of a low protein, medium carbohydrate and fat diet can ameliorate key PCOS reproductive traits. However, PCOS mice display a hindered ability for their metabolic system to respond to diet variations, and varying macronutrient balance did not have a beneficial effect on the development of metabolic PCOS traits. We reveal that PCOS traits in a hyperandrogenic PCOS mouse model are ameliorated selectively by diet, with reproductive traits displaying greater sensitivity than metabolic traits to dietary macronutrient balance. Hence, providing evidence to support the development of evidence-based dietary interventions as a promising strategy for the treatment of PCOS, especially reproductive traits.

[1] Discipline of Obstetrics and Gynaecology, School of Women's and Children's Health, University of New South Wales Sydney, Sydney, NSW 2052, Australia. [2] Charles Perkins Centre, University of Sydney, Sydney, NSW 2006, Australia. [3] Andrology Laboratory, ANZAC Research Institute, University of Sydney, Sydney, NSW 2139, Australia. [4] School of Medical Sciences, Department of Pathology, University of New South Wales Sydney, Sydney, NSW 2052, Australia. ✉email: k.walters@unsw.edu.au

Polycystic ovary syndrome (PCOS) is the most common endocrine disorder estimated to affect 8–13% of women of reproductive age worldwide[1–3]. PCOS is a heterogeneous disorder encompassing reproductive, metabolic and endocrine abnormalities, including hyperandrogenism, ovulatory disturbance, reduced fertility and a higher incidence of obesity, insulin resistance, hepatic steatosis, and dyslipidemia, which heighten the risk of type 2 diabetes and cardiovascular disease[1,4]. The prevalence of weight excess in PCOS is in the range of 30–75%[5,6], and obesity worsens symptoms of PCOS[7–9], while weight loss improves its features[5]. The 2018 international evidence-based guideline for the assessment and management of PCOS recommends that lifestyle intervention, including diet, exercise and behavioral strategies, should be implemented to all women with PCOS[10]. However, the optimal dietary composition to be incorporated into a lifestyle management plan aimed at improving the clinical features of PCOS remains unknown.

There is a paucity of well-controlled studies on dietary impacts on PCOS and further research is needed[5,8]. Studies within the limited literature available suggest that in women with PCOS, high fat diets aggravate obesity and PCOS traits[7]. Studies of the effects of low[11] and high protein[11–13], and low[14,15] and high carbohydrate[12] are also reported. Low carbohydrate diets are reported to exert greater reductions in insulin resistance and total cholesterol than standard diets[8], with an additional 1–5% significantly greater weight loss[14]. While a modified hypocaloric diet with a high-protein, low glycaemic load results in significantly increased insulin sensitivity[16], these studies also demonstrate that overall weight loss has a beneficial effect on PCOS features regardless of dietary composition. Indeed, a critical review of the literature identified that improvements in many key features of PCOS occur with a modest weight loss of 5–15%[17]. A recent systematic review assessing dietary effects on PCOS outcomes identified that the body of evidence was not sufficient to draw reliable conclusions about the optimal dietary composition for lifestyle management in PCOS[8].

Current guidelines recommend a 5–10% weight loss over 6 months in overweight women with PCOS as it improves clinical outcomes[10]. Nevertheless, energy-restricted dietary interventions have notoriously low long-term compliance. Moreover, weight management is difficult, with attrition a common problem, which appears to be even worse in PCOS patients, with a Cochrane review reporting attrition rates in lifestyle intervention studies of up to 46% in women with PCOS[18]. Furthermore, weight management interventions may be less effective in women with PCOS than in women without PCOS, based on the observation that longitudinal weight gain is greater in women with PCOS[19]. Despite their reported higher motivation to follow healthy weight management practices than women without PCOS, women with PCOS are still heavier[20]. This indicated change in response to diet may be explained by factors driving PCOS, such as hyperandrogenism and insulin resistance, which can alter energy homeostasis, appetite and/or metabolism, but this remains unclear.

Before an optimum dietary composition for the management of PCOS can be identified, the knowledge gap on the impact of macronutrient balance on PCOS features must be addressed. Rigorous and systematic research is needed to provide a basis for the development of testable nutritional strategies to be translated into human clinical studies[21]. However, human diet studies are commonly constrained by poor compliance beyond short-term trials; hence it is difficult to draw firm conclusions on optimum diets for sustained benefits. In contrast to human studies, identifying the principles of optimal macronutrient composition is more tractable in a rigorously controlled environment setting, such as in animal models. Carefully designed experiments using animal models that display characteristics of the human condition of PCOS with high fidelity are valuable tools as they afford insights into fundamental biological mechanisms impacting on the development of PCOS. To date, systematic experimental studies to elucidate the optimum diet for amelioration of PCOS traits have not been reported.

Previously, we combined the mouse as a mammalian model and the Geometric Framework (GF), a powerful nutritional approach, to analyze the individual and interactive effects of protein, carbohydrate and fat on reproductive and metabolic function[22,23]. We revealed that reproductive function and cardiometabolic health are strongly impacted by the macronutrient balance of the diet[23,24], therefore we hypothesized that dietary macronutrient balance will have a significant impact on the development of PCOS features. In this study, we combined a PCOS mouse model[25,26] with the GF to evaluate the effect of dietary macronutrients (protein, carbohydrate and fat) on traits of PCOS. Our findings reveal that the development of PCOS traits is influenced by dietary macronutrient balance. We show that the reproductive PCOS feature of ovulatory dysfunction can be rescued by an optimized dietary composition. Hyperandrogenic PCOS mice exhibit a limited ability for their metabolic system to respond to variations in diet, and alterations in dietary macronutrient balance have a minimal beneficial effect on ameliorating metabolic PCOS traits.

## Results

### DHT treatment validation and response surface interpretation.
We evaluated the effects of dietary macronutrient balance on the development of reproductive and metabolic features of PCOS in PCOS and control female mice with dihydrotestosterone (DHT)-induced experimental PCOS. DHT treatment was validated by liquid chromatography-tandem mass spectrometry (LC-MS/MS), with PCOS mice ($1.34 \pm 0.17$ ng/ml) exhibiting a significant increase in circulating DHT levels compared to control females ($0.27 \pm 0.04$ ng/ml, $P < 0.001$). Serum testosterone levels were also assessed and in agreement with our previous studies[25,26], levels were observed to be comparable between control ($0.07 \pm 0.01$ ng/ml) and PCOS ($0.05 \pm 0.02$ ng/ml) females. Using the GF nutritional modeling method, cohorts of control and DHT-induced PCOS mice were each confined to one of ten diets that varied systematically in protein (P), carbohydrate (C), and fat (F) content, which allowed us to plot a landscape profile of different PCOS phenotypes onto macronutrient intakes (Supplementary Fig. 1 and Supplementary Table 1).

Response surfaces to diet for the various features of PCOS assessed are presented in three graphs of 2D nutrient spaces, with each cut through the median of the third nutrient axis (value provided in parenthesis below the x-axis label). The 2D slices visualize the impact of all three macronutrient dimensions (protein, P; carbohydrate, C; fat, F represented on the x and y axis) on the outcomes being assessed. The outcome (e.g., corpora lutea number, Fig. 1e) is represented on the colored surface as a contour, with the number on the surface being the outcome value. To determine differences in the presence of PCOS-like traits in response to diet, 3D GAM response surfaces are provided showing the difference in response between control and PCOS mice to macronutrients. All surfaces are interpreted statistically using general additive statistical models (GAMs)[23,24] or mixture models (Sheffe's polynomials) and results are provided in Supplementary Tables 2 and 3. GAMs are a form of multiple regression, which allow for the quantification of non-linear effects of multi-dimensional predictors (e.g., dietary intake of P, C, and F) on response variables (e.g., metabolic/reproductive parameters) using non-parametric smoothing functions (here thin-plate and cubic-regression splines).

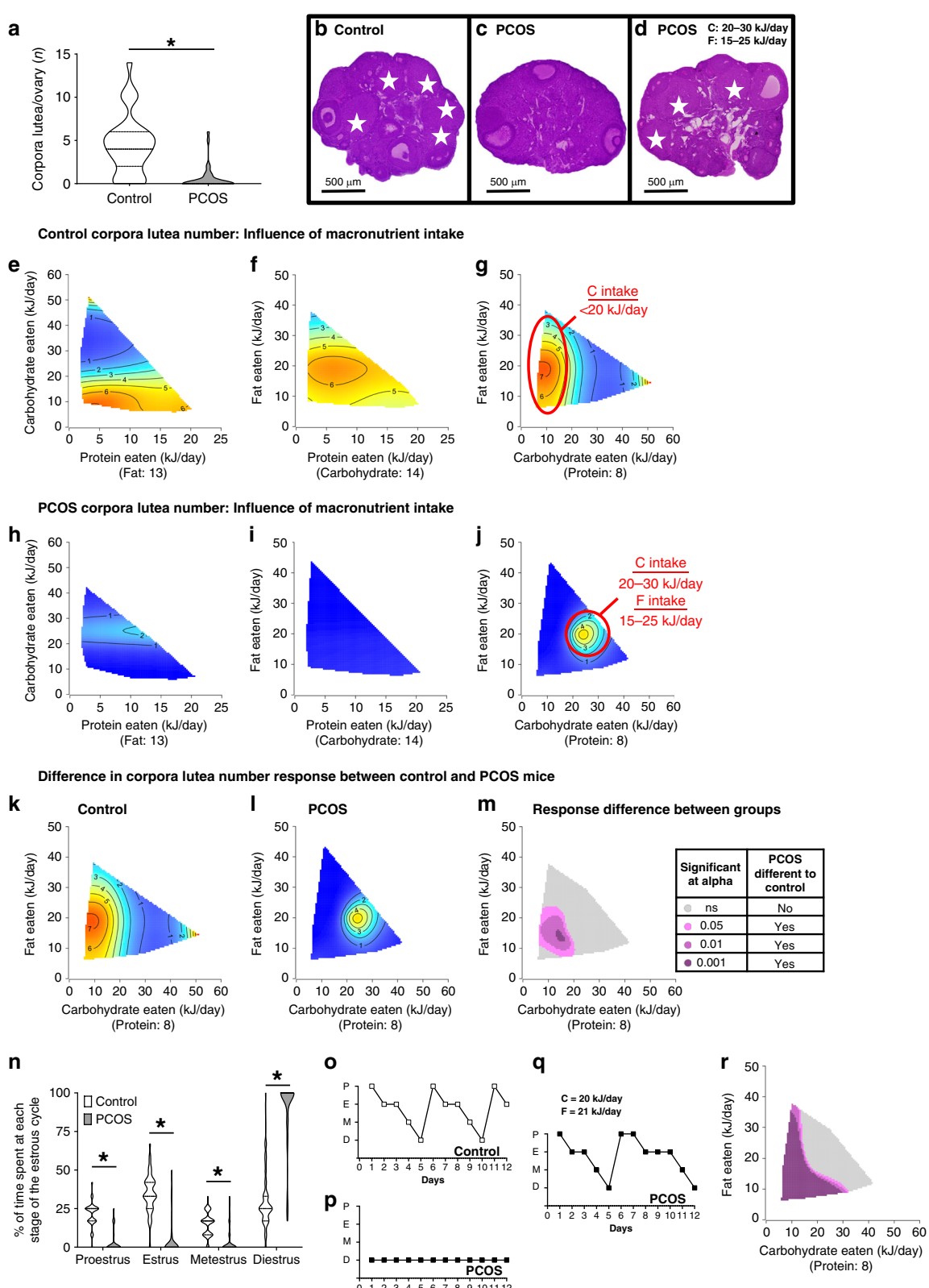

**a** Corpora lutea/ovary (n) — Control vs PCOS

**b** Control, **c** PCOS, **d** PCOS C: 20–30 kJ/day F: 15–25 kJ/day (scale bars 500 μm)

**Control corpora lutea number: Influence of macronutrient intake**

**e** Carbohydrate eaten (kJ/day) vs Protein eaten (kJ/day) (Fat: 13)

**f** Fat eaten (kJ/day) vs Protein eaten (kJ/day) (Carbohydrate: 14)

**g** Fat eaten (kJ/day) vs Carbohydrate eaten (kJ/day) (Protein: 8) — C intake <20 kJ/day

**PCOS corpora lutea number: Influence of macronutrient intake**

**h** Carbohydrate eaten (kJ/day) vs Protein eaten (kJ/day) (Fat: 13)

**i** Fat eaten (kJ/day) vs Protein eaten (kJ/day) (Carbohydrate: 14)

**j** Fat eaten (kJ/day) vs Carbohydrate eaten (kJ/day) (Protein: 8) — C intake 20–30 kJ/day, F intake 15–25 kJ/day

**Difference in corpora lutea number response between control and PCOS mice**

**k** Control, **l** PCOS, **m** Response difference between groups

| Significant at alpha | PCOS different to control |
| --- | --- |
| ns | No |
| 0.05 | Yes |
| 0.01 | Yes |
| 0.001 | Yes |

**n** % of time spent at each stage of the estrous cycle (Control vs PCOS) — Proestrus, Estrus, Metestrus, Diestrus

**o** Control (Days), **p** PCOS (Days), **q** C = 20 kJ/day, F = 21 kJ/day PCOS (Days), **r** Fat eaten (kJ/day) vs Carbohydrate eaten (kJ/day) (Protein: 8)

**Selective macronutrient intakes rescue ovulatory dysfunction.** Irregular menstrual cycles and ovulatory dysfunction are key diagnostic features of PCOS[10]. In this study, irrespective of diet and consistent with our previous studies[25,26], PCOS mice exhibited a significant disturbance of ovulatory function with a decrease in corpora lutea (CL) number (0.5 ± 0.2) compared to control mice (4.5 ± 0.6) (P < 0.001, Fig. 1a–c). However, CL were present within the ovaries of a subset (8/35) of PCOS mice (Fig. 1d). We used response surfaces to detail the effects of all three macronutrients on CL response. All control mice ovulated and displayed maximal CL numbers on dietary intakes of <20 kJ/day C and 10–25 kJ/day F, shown by the area circled in red (CxF P = 0.033; Fig. 1g and Supplementary Table 2). The majority (77%, 27/35) of PCOS females analyzed were anovulatory as

**Fig. 1 An optimal dietary macronutrient balance rescues ovulatory dysfunction in a PCOS mouse model. a** Number of corpora lutea (CL) in control and PCOS mice, showing that regardless of diet PCOS mice exhibit fewer CL ($P < 0.001$). Significance determined by Mann–Whitney test, asterisk (*) indicates $P$-value <0.05. **b–d** Representative histological control (**b**) and PCOS (**c**) ovarian cross-sections, showing presence of CL in ovaries from PCOS mice with a macronutrient intake of C 20–30 kJ/day and F 15-25 kJ/day (**d**). White stars, corpora lutea. Magnification ×4, scale bar 500 µm. **e–j** 3D GAM response surfaces showing the relationship between macronutrient intake (kJ/day) and number of CL in control (**e–g**) and PCOS ovaries (**h–j**) showing restoration of ovulations in PCOS mice on a macronutrient intake of C 20–30 kJ/day and F 15–25 kJ/day (**j**). Red areas indicate the greatest value, which then decreases to the lowest value as the color shifts to blue. Regions with highest CL number for control (**g**) and PCOS (**j**) mice are encircled in red. **k–m** 3D GAM response surfaces showing the impact of main drivers (carbohydrate and fat) on CL numbers in control mice (**k**), PCOS mice (**l**) and the response difference (**m**), demonstrating a comparable ovulatory response between control and PCOS above a C intake of >20 kJ/day and F > 25 kJ/day. **a, e–m** $n = 36$ control and 35 PCOS mice. **n** % of time spent at each stage of the estrous cycle for control and PCOS mice, showing that a subset of PCOS mice did cycle through all estrous cycle stages. Significance determined by Mann–Whitney test, asterisk (*) indicates $P$-value <0.05, all $P < 0.001$. **o–q** Representative graphs of estrous-cycle patterns, showing restoration of cyclicity in a PCOS mouse with a dietary intake of 20 kJ/day C and 21 kJ/day F (**q**). **r** Response difference between control and PCOS mice for number of estrous cycles completed, demonstrating comparable estrous cycle response between control and PCOS mice at similar intakes that restore ovulatory response. **n, r** $n = 93$ control and 94 PCOS mice. **a, n** Data presented as violin plots, doted lines indicate the median and dashed lines the 25 and 75% percentiles.

demonstrated by the lack of CL in their ovaries (Fig. 1c) and by the dark blue areas on the response surfaces (Fig. 1h–j). However, the response surface identified that CL were observed in PCOS mice that ingested a tightly defined combination of 20–30 kJ/day C and 15–25 kJ/day F, at a median of 8 kJ/day P, highlighted by the area circled in red (Fig. 1j). Response surfaces demonstrated that the combination of C and F intakes was the main driver of ovulation in control and PCOS mice (shown by red/yellow hot-spots in Fig. 1g, j, Supplementary Table 2). Therefore, we subtracted C-F response surfaces derived from control mice (Fig. 1k) from those generated from PCOS animals (Fig. 1l) in order to identify how control and PCOS mice differed in their response to diet (Fig. 1m). The ovulatory response to diet in control and PCOS mice was comparable when they ingested >20 kJ/day C and >25 kJ/day F (shown by non-significant gray area Fig. 1m). In contrast on intakes of <20 kJ/day C and <25 kJ/day F, control and PCOS mice had significantly different responses to diet (shown by pink-purple area Fig. 1m). These findings demonstrate that control mice are able to maintain ovulations on a wide range of dietary intakes, while PCOS mice are only able to ovulate under a narrower nutritional niche. In agreement with our previous studies[25,26], circulating FSH and LH levels were not altered between control and PCOS mice (Supplementary Fig. 2a and 3a) and both groups displayed similar responses to diet (Supplementary Fig. 2b-j and 3b-j). However, diet-induced restoration in ovulatory function in PCOS females was confirmed by examining estrous cycle patterns in control and PCOS mice (Fig. 1n-r and Supplementary Fig. 4). A proportion of PCOS mice cycled (15/94), demonstrated by the finding that some PCOS mice were observed to spend time at all stages of the estrous cycle (Fig. 1n). While the majority of PCOS mice were acyclic, corresponding with the macronutrient balance able to restore ovulatory function in PCOS mice, a subset of PCOS mice within the intake ranges of 20–30 kJ/day C and 15–25 kJ/day F displayed completed estrous cycles (Fig. 1q and Supplementary Fig. 4g).

**PCOS caused weight increase regardless of dietary intake.** Women with PCOS exhibit a higher rate of weight gain and a higher prevalence of obesity compared to women without PCOS[19,27]. Regardless of dietary intervention, PCOS mice displayed a significant increase in body weight (23.5 ± 0.3 g) compared to control mice (21.7 ± 0.2 g) ($P < 0.001$, Fig. 2a), which is in agreement with previous studies using our hyperandrogenized PCOS mouse model[25,26]. The increased weight gain in PCOS females was reflected by an increase in body fat content ($P < 0.001$) and also lean mass ($P < 0.001$) (Fig. 2b–e). Parametrial, retroperitoneal, mesenteric and brown fat depot weights were all significantly increased in PCOS mice compared to control mice

($P < 0.05$, Fig. 2c). Response surfaces show that all three macro-nutrients influenced weight gain and that there is a similar color shift from blue to red along the P, C, and F axis in both control and PCOS mice (Fig. 2f–k, Supplementary Table 2). These data imply that in both groups body weight follows the same pattern of response to macronutrient intake, with body weight increasing as caloric intake of P, C, and F increased (Fig. 2f–k). However, PCOS mice were observed to gain substantially more weight at lower caloric intakes than control mice. This is exemplified in PCOS mice on lower intakes for each macronutrient, as they were found to be significantly heavier than control mice with the same P, C, or F intakes (Fig. 2l–n). Moreover, analysis of changes in response to diet for body weight between control and PCOS mice confirmed that PCOS mice were more sensitive to macronutrient intake, with PCOS mice gaining significantly more weight compared to control mice at low C and low F intakes of <25 kJ/day each ($P < 0.05$, Fig. 2q).

To identify reasons for this observed selective increase in body weight in PCOS mice, we assessed total food and energy intake against dietary composition. Food and energy (food intake × energy (kJ) per gram P, C, and F in diet) intakes were plotted as response surfaces onto diet composition axis and interpreted statistically using mixture models (Scheffe's polynomials) (Fig. 3a–d, Supplementary Table 3). Both control and PCOS mice exhibited the same eating patterns with food and energy intake decreasing as protein content increased as expected, evidenced by the red to blue color shift along the P axis (Fig. 3a–d); ruling out increased food and energy intake as a potential cause for PCOS mice exhibiting an increase in weight gain. Instead, it implies a difference in post-ingestive responses to diet intake. In an additional dietary choice study, PCOS mice displayed no significant difference in their preference for P, C, or F compared to control mice (Fig. 3e). Furthermore, metabolic cage analysis of the control and PCOS mice, 1 week prior to the dietary choice study, consuming a standard chow diet revealed there was no significant difference between the two groups for day and night food intake, indirect calorimetry measurements of energy expenditure, body weight corrected energy expenditure, oxygen consumption ($vO_2$), carbon dioxide expulsion ($vCO_2$), respiratory exchange ratio (RER), and night locomotor activity (Table 1). Although PCOS mice exhibited lower day locomotor activity, this difference was relatively small and unlikely to be sufficient to fully explain the increase in body weight in PCOS mice. Hence, these data imply that although PCOS mice exhibited the same eating pattern, caloric intake, and activity as control mice, the underlying androgen excess driven pathogenesis of PCOS caused the PCOS mice to gain significantly more weight at intakes of <25 kJ/day C and <25 kJ/day F (Fig. 2q). However,

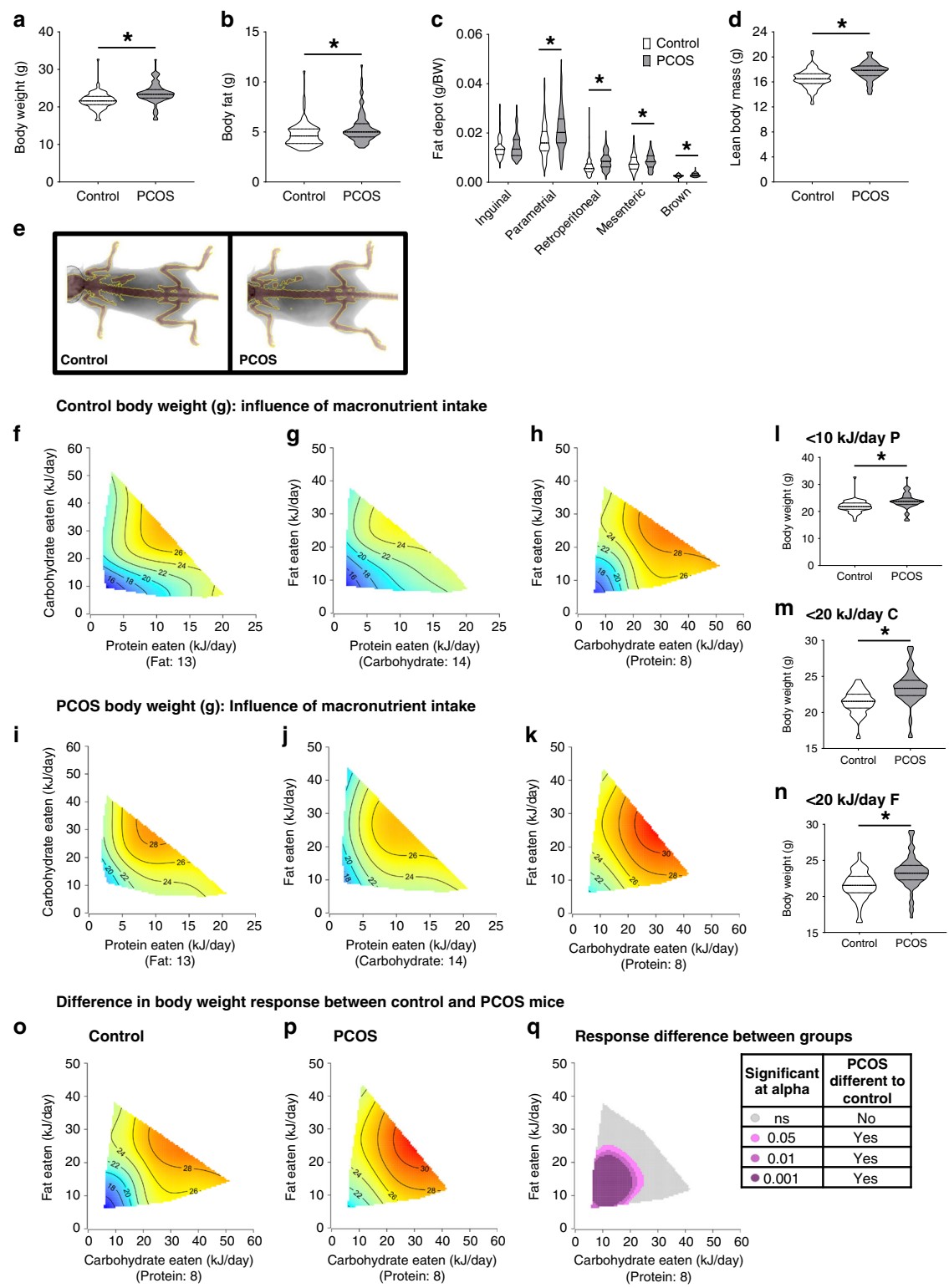

when C and F intakes were more than 25 kJ/day, body weight, although still greater in PCOS mice, was influenced similarly by macronutrient intake in PCOS and control mice (Fig. 2q).

**Diet variations do not impact PCOS traits in adipose tissue.** When data across all diets was combined, hyperandrogenism induced a significant increase in adipocyte area within PCOS parametrial fat pads compared to control mice ($1729 \pm 54.3 \, \mu m^2$ vs. $1573 \pm 50.6 \, \mu m^2$) ($P = 0.04$, Fig. 4a, b). This finding is in

agreement with previous mouse PCOS studies[26,28]. Analysis of changes in response to diet for adipocyte size, revealed the same pattern as changes in response to diet for body weight between control and PCOS mice. PCOS mice were more sensitive to low macronutrient intakes, but otherwise both groups displayed comparable patterns of response to macronutrient intake (Fig. 4c). Adiponectin is an adipokine that is reduced in serum of PCOS patients[29–31] and PCOS animal models[26,30]. In this study, adiponectin levels were significantly decreased in PCOS females compared to control females ($11221 \pm 453.8$ ng/ml vs. $16806 \pm$

**Fig. 2 PCOS mice gain excess weight regardless of macronutrient balance. a** Body weight; regardless of diet PCOS mice exhibit increased body weight ($P < 0.001$). **b** Body fat (g), calculated by dual-energy x-ray absorptiometry (DEXA), showing that PCOS mice display a significant increase in body fat ($P = 0.0005$). **c** Fat depot weights showing a significant increase in adiposity in PCOS mice ($P < 0.05$). **d** Lean body mass (g) calculated by DEXA, showing that PCOS mice exhibit a significant increase in lean body mass ($P < 0.001$). **a–d** Significance determined by Mann–Whitney test, asterisk (*) indicates $P$-value <0.05. **e** Representative DEXA images of a control and PCOS mouse, showing development of increased adiposity in PCOS mouse. **f–k** 3D GAM response surfaces showing the relationship between macronutrient intake (kJ/day) and body weight in control (**f–h**) and PCOS (**i–k**) mice. Red areas indicate the greatest value, which decreases to the lowest value as the color shifts to blue. Response surfaces show that dietary macronutrient intake influenced body weight similarly in control and PCOS mice, as color shifts from blue to red along the P, C, and F axis. Direct comparison of control and PCOS mice on <10 kJ/day P (**l**), $n = 56$ control and 62 PCOS mice, <20 kJ/day C (**m**), $n = 74$ control and 68 PCOS mice and <20 kJ/day F intakes (**n**), $n = 76$ control and 70 PCOS mice, demonstrate that PCOS mice are significantly heavier than control mice on similar P, C, and F intakes ($P < 0.001$). **o–q** 3D GAM response surfaces showing the effects of C and F intakes on body weight in control mice (**o**), PCOS mice (**p**) and the response difference (**q**), revealing PCOS mice exhibited a significantly different (higher) body weight compared to control mice at C and F intake <25 kJ/day. Above a C and F intake of 25 kJ/day, PCOS mice exhibit a comparable body weight response to macronutrient intake as controls. **a–d, f–k, o–q**, $n = 93$ control and 94 PCOS mice. **a–d, l–n** Data presented as violin plots, doted lines indicate the median and dashed lines the 25 and 75% percentiles.

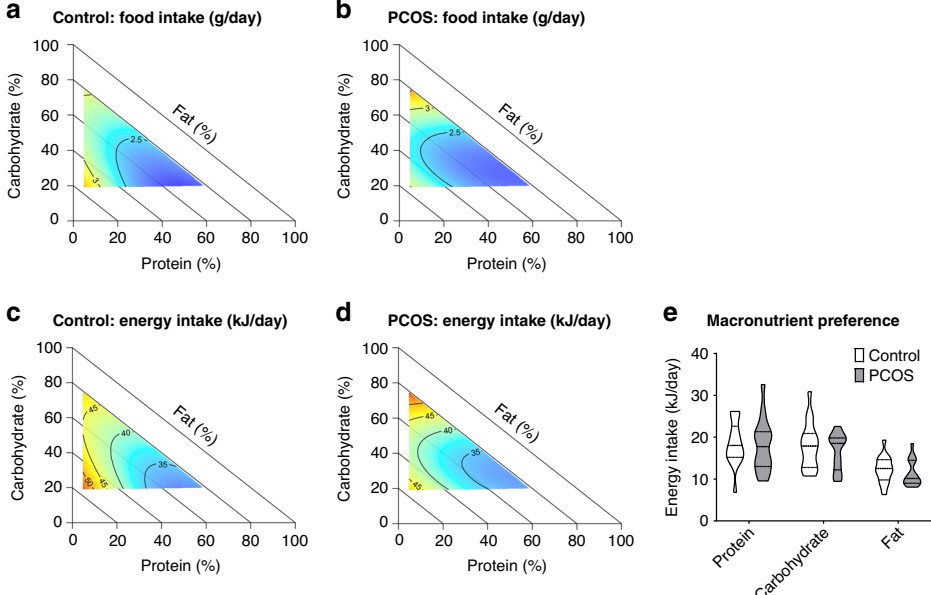

**Fig. 3 PCOS and control mice exhibit similar eating habits and dietary preference.** Response surfaces showing food intake of control (**a**) and PCOS (**b**) mice and energy intake of control (**c**) and PCOS (**d**) mice across 10 different diets varying in protein (P), carbohydrate (C), and fat (F) content, demonstrating that PCOS mice exhibit the same eating pattern as control mice, as dietary protein increases, food and energy intake decreases. Red areas indicate the greatest value for each response, which then decreases to the lowest value as the color shifts to blue. **a–d** $n = 93$ control and 94 PCOS mice. **e** Macronutrient intake (kJ/day) of control and PCOS mice when given a choice between 3 diets (high P (60%), high C (75%), and high F (75%)), showing no significant difference in macronutrient preference between control and PCOS mice. **e** $n = 17$ control and 16 PCOS mice. **e** Data presented as violin plots, doted lines indicate the median and dashed lines the 25 and 75% percentiles.

758.7 ng/ml) ($P < 0.001$, Fig. 4d). In response to diet, adiponectin levels most strongly increased with increasing C intake in control mice ($P = 0.002$), however P intake also exerted an influence over adiponectin levels ($P = 0.032$, Fig. 4e, g, Supplementary Table 2). In contrast, in PCOS mice macronutrient intake had no significant influence on adiponectin levels, shown by the monotone blue response surfaces across all diets (Fig. 4h–j). Hence, revealing that diet was not able to modify the androgen excess-driven aberrant adipocyte function in PCOS mice. This difference in response is clearly visualized after the subtraction of C–F response surfaces of control (Fig. 4k) and PCOS mice (Fig. 4l), where the majority of control and PCOS mice exhibit a significantly different response to macronutrient intake for serum adiponectin levels (shown by purple in Fig. 4m).

**Cholesterol in PCOS mice is minimally impacted by diet.** Regardless of dietary intervention, there was a trend for PCOS mice to exhibit an increase in serum cholesterol levels

(61.5 ± 2.7 mg/dL) compared to controls (53.1 ± 2.2 mg/dL, $P = 0.06$; Fig. 5a). Response surfaces revealed that control mice were readily able to regulate serum cholesterol levels on a wide range of macronutrient intakes, as control mice displayed consistent, low levels of circulating cholesterol, as shown by the overall hue of blue (Fig. 5b–d). In contrast, serum cholesterol levels in PCOS mice was highly affected by F intake, with increasing F consumption leading to increased serum cholesterol levels, as seen by the change of color from blue to red along the F axis ($P < 0.001$, Fig. 5f–g, Supplementary Table 2). The subtraction of serum cholesterol C–F response surfaces of control mice (Fig. 5h) and PCOS mice (Fig. 5i) showed largely a significant difference in response (Fig. 5j) to diet between the two groups. This finding revealed that overall macronutrient intake minimally influenced cholesterol levels in control mice, and implies that the underlying androgen excess driven pathogenesis of PCOS caused the PCOS mice to have a reduced ability to regulate their cholesterol levels in response to changes in macronutrient intake. Irrespective of

**Table 1 Metabolic state does not differ between control and PCOS mice.**

| | Control | PCOS | Significance |
|---|---|---|---|
| Day | | | |
| Food intake (g) | 0.56 ± 0.06 | 0.49 ± 0.06 | Ns |
| Locomotor activity (beam breaks) | 14180 ± 662.8 | 12291 ± 552.9 | 0.037 |
| Energy expenditure (kcal/12 h) | 50.6 ± 2.6 | 51.5 ± 3.3 | Ns |
| Energy expenditure (kcal/12 h/kg) | 2144 ± 107.5 | 2019 ± 98.23 | Ns |
| $VO_2$ (ml/12 h/kg) | 7448 ± 371.4 | 6988 ± 338.8 | Ns |
| $VCO_2$ (ml/12 h/kg) | 5764 ± 301.9 | 5521 ± 275.3 | Ns |
| RER ($VCO_2/VO_2$) | 0.77 ± 0.01 | 0.79 ± 0.01 | Ns |
| | **Control** | **PCOS** | **Significance** |
| Night | | | |
| Food intake (g) | 2.26 ± 0.13 | 2.38 ± 0.13 | Ns |
| Locomotor activity (beam breaks) | 45748 ± 1660 | 46087 ± 2219 | Ns |
| Energy expenditure (kcal/12 h) | 70.3 ± 3.3 | 75.6 ± 3.2 | Ns |
| Energy expenditure (kcal/12 h/kg) | 2960 ± 139.70 | 2957 ± 99.69 | Ns |
| $VO_2$ (ml/12 h/kg) | 10071 ± 469.7 | 10045 ± 348 | Ns |
| $VCO_2$ (ml/12 h/kg) | 8723 ± 435.1 | 8763 ± 267.3 | Ns |
| RER ($VCO_2/VO_2$) | 0.85 ± 0.01 | 0.87 ± 0.01 | Ns |

Measurements of indirect calorimetry by metabolic cages of control and PCOS mice on a standard chow diet, showing food intake (g), locomotor activity (beam breaks), energy expenditure (kcal/12 h), body weight corrected energy expenditure (kcal/12 h/kg), oxygen ($O_2$) consumption ($VO_2$), carbon dioxide ($CO_2$) produced ($VCO_2$), and respiratory exchange ratio (RER) do not differ between control and PCOS mice. Data are the mean ± SEM; $n = 17$ control and 16 PCOS mice.

diet and consistent with our previous study[26] PCOS mice did not display a significant increase in serum triglyceride levels (Supplementary Fig. 5a). Triglycerides levels in control and PCOS mice were driven by P intake ($P < 0.05$, Supplementary Fig. 5b-d), and P and C intakes ($P < 0.05$, Supplementary Fig. 5e-g), respectively, but overall levels were within a similar range for both groups (40–65 mg/dL). In line with this, analysis of changes in response to diet between control and PCOS mice showed overall comparable responses to diet between both treatment groups (Supplementary Fig. 5j).

**Fasting glucose in PCOS mice is minimally affected by diet.** Women with PCOS often display elevated fasting glucose levels due to insulin resistance[32]. Irrespective of dietary intervention, fasting glucose levels were significantly elevated in PCOS mice compared to control mice (9.2 ± 0.2 mmol/L vs. 8.8 ± 0.2 mmol/L, $P < 0.05$; Fig. 6a), consistent with previous findings in our hyperandrogenized PCOS mouse model[26]. Response surfaces indicated that control mice exhibited superior stability over fasting glucose levels on different macronutrient intakes, with the majority of mice exhibiting glucose levels of ~8.5 mmol/L, observed in green-yellow (Fig. 6b–d). In contrast, PCOS mice displayed impaired control over fasting glucose levels with the majority exhibiting levels of 9.5 mmol/L or above across varying macronutrient intakes (Fig. 6e–g). Specifically, PCOS mice were influenced most by P and F intake, with increased P leading to the highest levels of fasting glucose in this group (Fig. 6e–f, Supplementary Table 2). Subtraction of C–F response surfaces of control (Fig. 6h) and PCOS mice (Fig. 6i), showed that the majority of PCOS mice on F intakes >12 kJ/day were unable to regulate serum fasting glucose to comparable levels observed in control mice (Fig. 6j). This result revealed that overall variations in macronutrient intake had a limited influence on fasting glucose levels in control mice; however, the presence of androgen excess-driven PCOS pathology significantly reduced the capability of PCOS mice to modulate glucose levels in response to diet. GTT response was also assessed and irrespective of diet and consistent with our previous studies[25,26], PCOS mice did not display an altered GTT response to control mice (Supplementary Fig. 6a), with both groups displaying similar GTT AUC across the

majority of dietary intakes (Supplementary Fig. 6b–g). Correspondingly, analysis of changes in response to diet between control and PCOS mice identified no difference in response to diet between both groups for GTT response (Supplementary Fig. 6h–j).

**Discussion**

Nutritional manipulations are one of the most widely-applicable and practical interventions shown to impact health in a wide range of mammals[33]. Hence, defining the optimal, balanced macronutrient dietary treatment for PCOS is of great practical interest, as dietary interventions would be an extremely appealing therapeutic approach to clinically manage PCOS pathology. However, this is challenging to achieve in humans due to the logistical challenges of diet design and compliance over prolonged periods required for sustained weight loss.

In this study, we utilized a PCOS mouse model that replicates numerous reproductive and metabolic features of human PCOS to gain insight into the basic principles on the individual and interactive effects of protein, carbohydrate and fat on the development of PCOS features. By combining an experimental PCOS mouse model with the GF nutritional modeling approach, we provide evidence that macronutrient balance is an important factor in the development of PCOS traits. Our data show that the PCOS traits of acyclicity and anovulation can be rescued by a specific dietary macronutrient balance, supporting the use of dietary manipulation as a potential viable strategy to be used to assist in restoration of ovulatory dysfunction in women with PCOS. In agreement with previous studies[25,26,34], ovulatory function is aberrant in the hyperandrogenic mouse model of PCOS as corpora lutea populations are diminished. However, ovulatory function was restored in PCOS females on a tightly defined intake range of protein, carbohydrate and fat, indicated by the presence of corpora lutea in their ovaries (8/35), which conclusively confirms the occurrence of recent ovulations. This restoration in ovulatory function in PCOS females was confirmed by the presence of complete estrous cycles in PCOS mice (15/94) on the same dietary macronutrient balance able to restore ovulation.

This finding reinforces clinical evidence supporting the use of lifestyle interventions in the treatment of dysfunctional menstrual

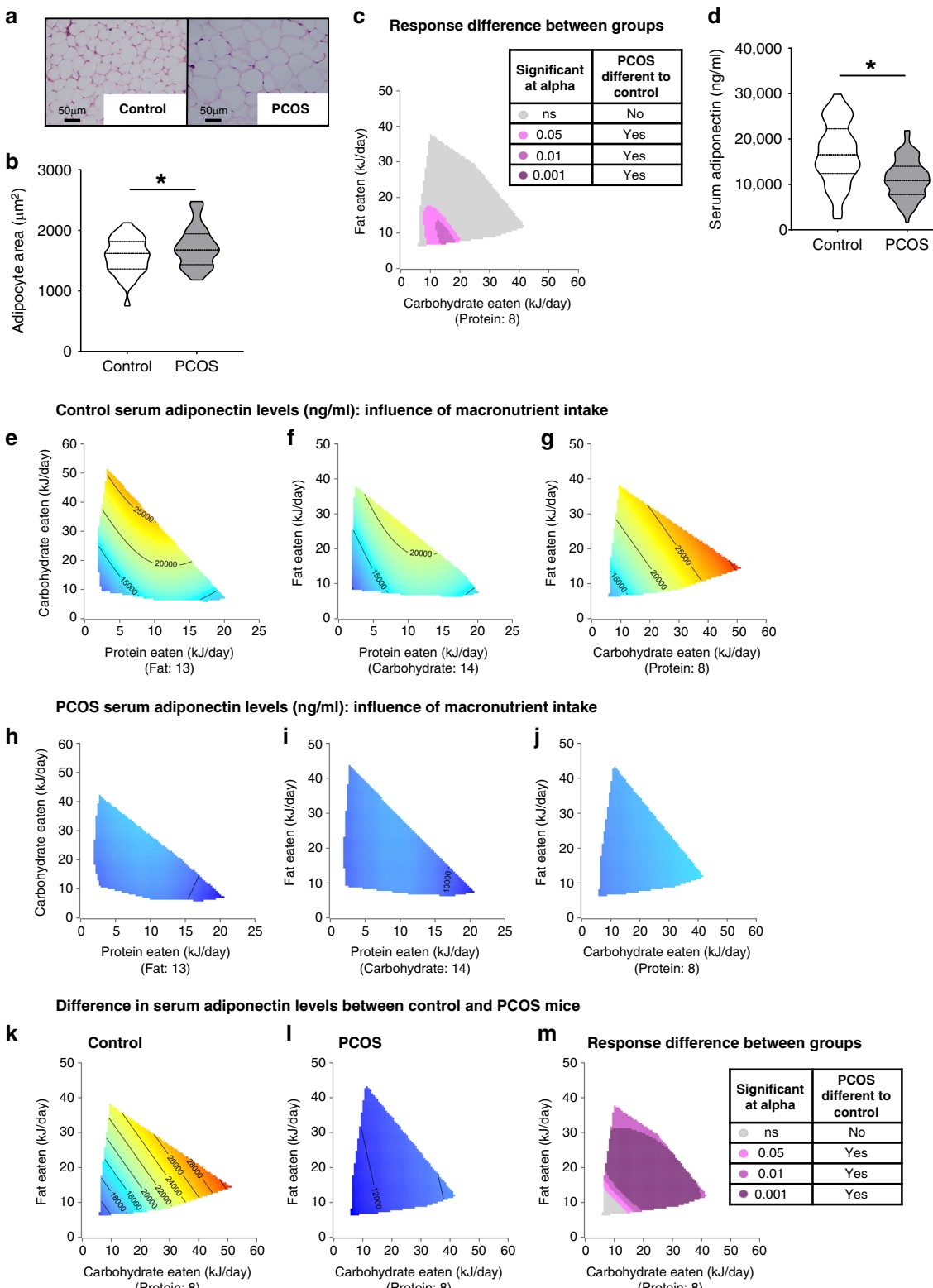

cycling and ovulation for infertile women with PCOS. In a previous study comparing the use of medical treatment (clomiphene citrate, metformin) vs. lifestyle modification (reduction in energy intake and an increase in physical activity) in patients with PCOS, it was observed that lifestyle modification effectively improved rates of menstrual cycles and clinical pregnancies to levels comparable with those reported in the clomiphene citrate and metformin treated women[35]. Furthermore, in another study using a structured exercise training and hypocaloric dietary intervention alone or together with clomiphene citrate in overweight women with PCOS, the lifestyle intervention alone significantly improved the metabolic and endocrine profile, and ovulation rate when in combination with clomiphene citrate[36]. However, these studies do not take into account macronutrient balance and at present there is a paucity of high quality literature in this area, hindering the development of testable nutritional strategies to be translated

**Fig. 4 Macronutrient balance does not modulate adipocyte hypertrophy or elevated adiponectin levels in a PCOS mouse model. a** Histological sections of representative parametrial fat pads from a control and PCOS mouse, showing development of adipocyte hypertrophy only in PCOS mice. Magnification ×40, scale bar 50 μm. **b** Adipocyte area, showing that regardless of diet PCOS mice exhibit adipocyte hypertrophy ($P = 0.04$). Significance determined by two-tailed Student's $t$ test, asterisk (*) indicates $P$-value <0.05. **c** Response difference between control and PCOS mice for adipocyte area, showing that adipocyte size in PCOS mice was significantly increased at low carbohydrate and fat intakes. **b–c** $n = 35$ control and 38 PCOS mice. **d** Serum levels of adiponectin, showing that irrespective of diet PCOS mice exhibit lower serum adiponectin levels ($P < 0.001$). Significance determined by two-tailed Student's $t$ test, asterisk (*) indicates $P$-value <0.05. **e–j** 3D GAM response surfaces displaying the relationship between macronutrient intake (kJ/day) and serum adiponectin levels in control (**e–g**) and PCOS mice (**h-j**), showing a global decrease of adiponectin in PCOS mice irrespective of macronutrient intakes. Red areas indicate the greatest value for each response, which then decreases to the lowest value as the color shifts to blue. **k–m** Response surfaces showing the effects of C and F intakes on adiponectin levels in control mice (**k**), PCOS mice (**l**), and the response difference (**m**), demonstrating that overall adiponectin levels are significantly different between control and PCOS mice, consequently serum adiponectin levels in PCOS mice are heavily influenced by the PCOS pathophysiology and not diet. **d–m** $n = 78$ control and 87 PCOS mice. **b**, **d** Data presented as violin plots, doted lines indicate the median and dashed lines the 25 and 75% percentiles.

into human clinical studies. The results of this study demonstrate that macronutrient balance does impact the development of reproductive PCOS traits and that an optimal dietary macronutrient balance of low protein and medium carbohydrate and fat dietary consumption can rescue ovulatory dysfunction observed in a PCOS mouse model. These findings provide support for the future development of an evidence-based nutritional strategy for the treatment of PCOS reproductive traits. Notably, the dietary macronutrient ratio identified (14% P, 47% C, and 39% F) falls within the ranges of a traditional Mediterranean diet, which has been associated with numerous health benefits such as improvement of lipid profile, protection against oxidative stress[37,38], decreased adiposity[39] and decreased risk of type-2 diabetes[40]. A recent study has reported that women with PCOS with a high adherence to a Mediterranean diet presented with lower testosterone and HOMA-IR levels[41]. Thus, further studies defining if this beneficial effect on PCOS traits is attributed to the macronutrient balance alone or the food choices associated with a Mediterranean diet are warranted.

Obesity is common in women with PCOS[5,6] and obesity itself worsens symptoms of PCOS[7–9]. Consistent with this, research using animal models has identified that hyperandrogenism leads to increased body weight in PCOS animal models[42] and high fat and western-style diets exaggerate androgen excess-induced metabolic features of PCOS[43–45]. Findings from this study have revealed that while protein, carbohydrate and fat intake are all key drivers of weight gain in both control and PCOS mice, PCOS mice displayed an increased sensitivity in response to diet compared to controls, as PCOS females gained more weight at lower macronutrient intakes than control mice. Congruent with our research findings, in a comprehensive review on the effect of different dietary compositions on outcomes in women with PCOS, only subtle differences were found between diets[8,46]. It was concluded that from the current information available, the best approach for a clinical benefit in overweight women with PCOS was a diet aimed at reducing body weight, regardless of dietary composition[8]. However, it was also noted that limited evidence is currently available and further research is required from high quality long-term RCTs assessing a range of dietary compositions[8].

Women with PCOS have been described as having significantly poorer dietary habits characterized by a greater consumption of high glycaemic index foods[47,48] and a decreased basal metabolic rate[49]. A significantly higher intake of energy from carbohydrates and a corresponding higher respiratory exchange ratio has also been reported in women with PCOS[50]. On the other hand, other studies have reported that women with and without PCOS display similar dietary habits and energy intake from protein, carbohydrate and fat consumption[51,52]. In fact in one study comparing women with and without a self-reported PCOS diagnosis, while women with PCOS reported a higher energy

intake, there was no difference in protein, carbohydrate or fat percent energy intake and actually better dietary intake in terms of diet quality, micronutrient, saturated fat and glycaemic index intake[53]. In our study we observed the same pattern for energy intake and no significant difference in macronutrient preference between control and PCOS mice. Moreover, evaluation of metabolic state in control and PCOS mice, using metabolic cages, revealed that overall metabolic measurements, including energy expenditure and RER, did not differ significantly between groups. This finding is congruent with the reported observation that women with PCOS do not have a different resting metabolic rate to women without PCOS[50]. Hence, taken together these findings infer that hyperandrogenism does not drive notable changes in eating habits, dietary preference or sedentary activity, but rather that hyperandrogenism decreases the capacity of the metabolic system to cope with poorer diets.

Adiponectin, a protein hormone secreted by adipocytes, has been reported to be reduced in women with PCOS[31,54,55], and insufficient levels linked with the development of insulin resistance in murine models of obesity[56]. Overexpression of adiponectin in an androgen excess-induced mouse model of PCOS was discovered to maintain healthy metabolic parameters of PCOS, while in contrast a loss of adiponectin caused DHT-induced PCOS mice to become even more insulin resistant than their control DHT-exposed littermates[28]. Furthermore, exogenous administration of adiponectin rescues reproductive and metabolic PCOS-like traits in two different rodent models of PCOS[30,57]. In this study, PCOS females exhibited an increase in adipocyte size compared to controls, with altered function evident by a significant decrease in serum adiponectin levels. These data are in agreement with our and other animal models of PCOS[26,28,57] and previous clinical reports stating that women with PCOS tend to have low levels of adiponectin compared to women without PCOS[31,54,55]. Moreover, the current data demonstrate that alterations in dietary macronutrient balance were unable to beneficially influence adiponectin levels in PCOS mice. It has been reported that women with PCOS exposed to either a hypocaloric diet with a standard macronutrient balance or a hypocaloric diet with a low carbohydrate and high protein balance both displayed an increase in adiponectin levels. The authors concluded the increase in adiponectin was most likely due to weight loss rather than the alteration in macronutrient balance[16]. Our findings agree with this as we show that different macronutrient ratios were not able to significantly change adiponectin levels, indicating that diet is not a key influence on the management of adiponectin levels in a PCOS population. Instead, we propose that hyperandrogenism is at the root of adipose tissue dysfunction, which is supported by the finding that normal serum adiponectin levels are maintained by a global loss of androgen receptor function in a hyperandrogenized PCOS mouse model[26].

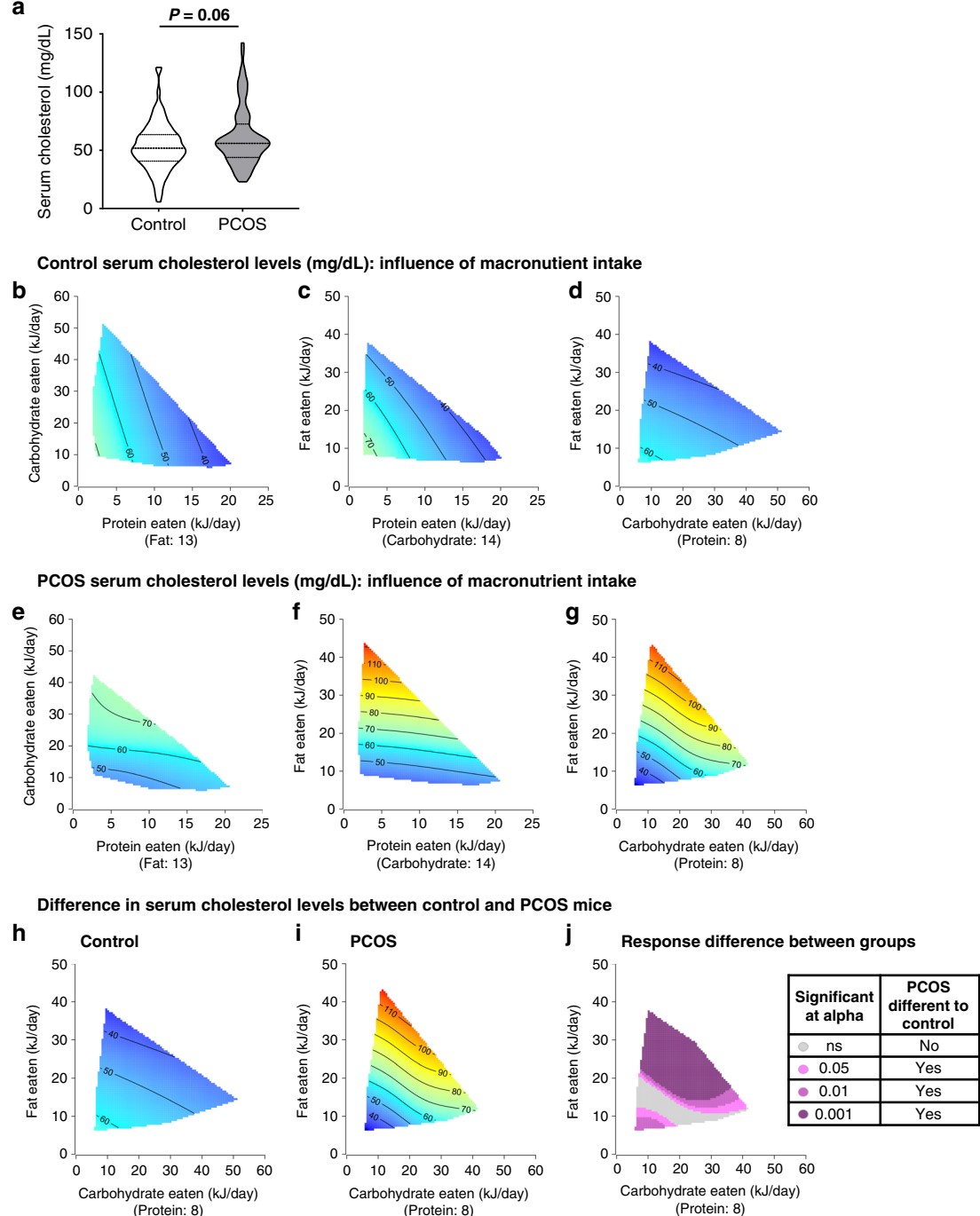

**Fig. 5 Increased cholesterol level in PCOS mice is driven by PCOS pathophysiology and cannot be modulated via diet. a** Serum cholesterol levels, showing that regardless of diet androgen excess leads to a non-significant increase in serum cholesterol levels ($P = 0.06$). Significance determined by Mann–Whitney test. Data presented as violin plots, doted lines indicate the median and dashed lines the 25 and 75% percentiles. **b–g** 3D GAM Response surfaces displaying the relationship between macronutrient intake (kJ/day) and serum cholesterol levels in control (**b–d**) and PCOS mice (**e–g**), showing that macronutrient intake had no influence on cholesterol levels in control mice, while F intake was the main driver for increased serum cholesterol levels in PCOS mice (**f, g**). Red areas indicate the greatest value for each response, which then decreases to the lowest value as the color shifts to blue. **h–j** 3D GAM response surfaces showing the effects of C and F intakes on serum cholesterol levels in control mice (**h**), PCOS mice (**i**) and response difference (**j**), showing that overall cholesterol levels in PCOS mice are significantly different to control mice due to the underlying pathophysiology of PCOS. However, serum cholesterol levels in PCOS females reached comparable levels to control when F intake was reduced to <20 kJ/day. **a–j** $n = 91$ control and 89 PCOS mice.

Dyslipidemia, a risk factor for cardiovascular disease, is prevalent in women with PCOS[58]. A meta-analysis showed that in an age-matched population of women with and without PCOS, women with PCOS displayed higher levels of LDL-cholesterol and non HDL-cholesterol, and that even when matched for BMI,

women with PCOS still exhibited higher LDL-cholesterol levels than controls[59]. Lifestyle modification improves the lipid profile in PCOS women[35] and improvements of cholesterol levels are observed under hypocaloric conditions regardless of diet composition[12,16]. Interestingly, in a study comparing the effects

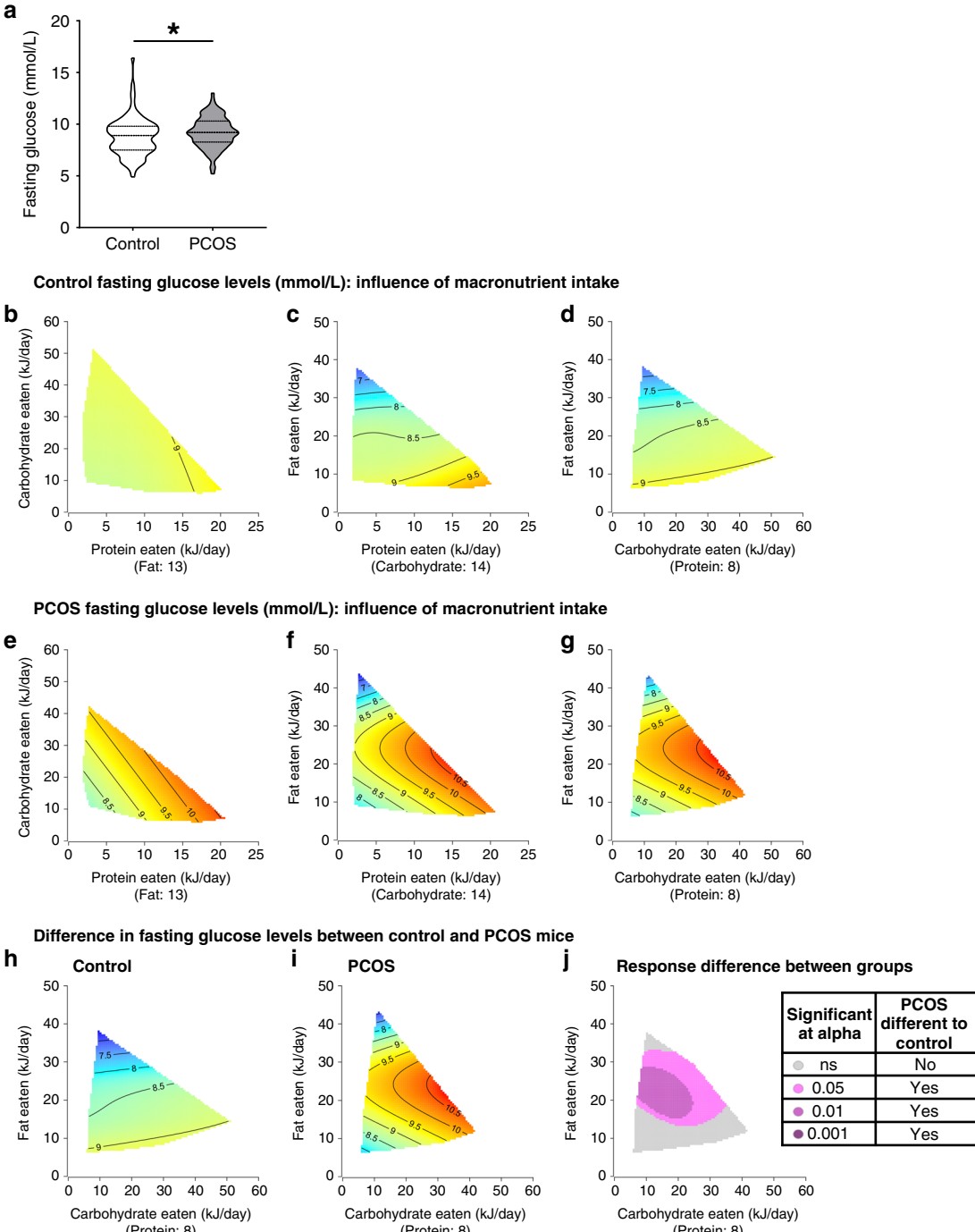

**Fig. 6 Increased fasting glucose levels in PCOS overall cannot be ameliorated by dietary interventions. a** Fasting glucose levels, showing a significant increase in basal glucose levels in PCOS mice compared to control mice ($P = 0.04$). Significance determined by Mann–Whitney test, asterisk (*) indicates $P$-value <0.05. Data presented as violin plots, doted lines indicate the median and dashed lines the 25 and 75% percentiles. **b–g** 3D GAM response surfaces displaying the relationship between macronutrient intake (kJ/day) and glucose levels in control (**b–d**) and PCOS mice (**e–g**). Red areas indicate the greatest value for each response, which then decreases to the lowest value as the color shifts to blue. Control mice exhibited a decrease in fasting glucose with increasing fat intake while PCOS mice display overall increased levels of fasting glucose. **h–j** Response surfaces showing the effects of C and F intakes on fasting glucose levels in control mice (**h**), PCOS mice (**i**) and response difference (**j**), demonstrating that fasting glucose levels in PCOS mice are comparable to control mice when fat intake was reduced to <12 kJ/day. On the other hand, fasting glucose is not able to be modulated by diet when PCOS mice ingested F in the range of 12–32 kJ/day, indicating that PCOS pathology hinders serum fasting glucose levels in PCOS mice with higher F intakes. **a–j** $n = 93$ control and 94 PCOS mice.

of either a therapeutic lifestyle changes diet or a pulse-based diet in women with PCOS, those on the pulse diet had a greater decrease in cardio-metabolic risk factors such as LDL-cholesterol and insulin response to OGTT[60]. As previously reported[25,26], our

PCOS mouse model mimics PCOS associated dyslipidemia, with an increase in serum cholesterol levels. In this study, control mice were readily able to regulate their serum cholesterol levels over a wide range of macronutrient intakes. In contrast, PCOS mice

displayed an impaired ability to regulate their cholesterol levels in response to changes in macronutrient intake, and serum cholesterol levels was highly affected by fat intake, with increasing fat consumption leading to increased serum cholesterol levels. Therefore, we propose that the hyperandrogenic PCOS environment is the key cause of dyslipidemia in PCOS females. This notion is supported by other animal studies where a global and neuron-specific loss of androgen receptor signaling prevented androgen excess-induced dyslipidemia[26]. In terms of macronutrients, decreasing dietary fat intake led to a decrease in serum cholesterol levels in PCOS females. This is in agreement with previous studies, where exposing PCOS mice to a high fat diet exacerbated the metabolic phenotype of PCOS[44].

The pattern of PCOS females being unable to regulate adiponectin and cholesterol across various diets, also extended to their inability to regulate serum glucose levels. In our study, as per our previous publications[26,34], impaired glucose homeostasis was observed in PCOS females. This finding is consistent with studies in humans where women with PCOS have increased levels of steady state plasma glucose and fasting glucose compared to non-PCOS women[32]. A lifestyle intervention combining exercise and a hypocaloric diet in women with PCOS reduced fasting insulin levels and HOMA-IR, but not fasting glucose levels[36]. In this study, fasting glucose levels were significantly elevated in PCOS mice compared to control mice, and overall glucose levels were unable to be restored to control levels by the wide range of dietary modifications. Control mice exhibited superior stability over fasting glucose levels on a wide range of macronutrient intakes, while PCOS mice displayed brittle control over fasting glucose levels. This finding implies that, as was the case with adiponectin and cholesterol regulation, the presence of androgen excess driven PCOS pathology significantly reduced the capability of PCOS mice to modulate glucose levels in response to varying diets.

The identification of an optimal dietary intervention for PCOS is of high importance as it would provide a widely feasible and cost-effective nutritional strategy for women worldwide suffering from PCOS. By combining the multidimensional nutritional GF with a PCOS mouse model, we have shed light on the impact of all three macronutrients, protein, carbohydrate and fat, on the development of different PCOS features. We reveal that ovulatory dysfunction and acyclicity can be rescued in our PCOS mouse model through the use of an optimal dietary intervention, indicating that dietary macronutrient composition should be considered when planning an optimal diet to assist in the management of PCOS-induced infertility. The finding that PCOS mice gained more weight at lower macronutrient intakes than control mice, supports implementation of the current clinical recommendations of weight loss and exercise for all women with PCOS. Hyperandrogenic PCOS mice also displayed a decreased capacity for their metabolic system to respond to variations in diet, and changes in macronutrient balance had a minimal beneficial effect on ameliorating metabolic dysfunction observed in PCOS mice. These findings support the notion that the underlying pathophysiology of PCOS has a predominant influence on the development of PCOS traits. Overall, we provide insight into the principles of macronutrient balance on PCOS traits, and strong evidence to support a place for the use of dietary interventions in the management of PCOS, in particular future development of evidence-based nutritional strategies for the treatment of PCOS reproductive traits.

## Methods

**Mice and diets**. C57Bl/6J female mice were housed in groups of 2 or 3 mice per cage (4–5 cages per diet) and maintained under standard housing conditions (ad libitum access to food and water in a temperature-controlled and humidity-controlled, 12-h light/dark environment) at the ANZAC Research Institute.

Specifically, room temperature set point is 22 +/− 1 °C and room humidity is between 60 and 70%. At 7 weeks of age, control and DHT-induced PCOS mice were switched from standard chow diet and provided ad libitum access to one of 10 experimental diets varying in protein, carbohydrate, and fat content (Specialty Feeds). The selection of macronutrient content used in the experimental diets was chosen, based on our previous publications[23,24], for optimal power in fitting response surface models used for statistical analysis (Supplementary Fig. 1 and Supplementary Table 1). Ten control mice/diet were provided ad libitum access to diets 1, 2, 4–10. 10 PCOS mice/diet were provided ad libitum access to diets 1, 2, 4–9, and 11 mice for diet 10. Diet 3 was discontinued after 6 control and 6 PCOS mice experienced a weight loss of ≥20% or failed to thrive. Hence, data from diet 3 is from 3 control and 3 PCOS mice that completed the 10 weeks of experimental diet exposure. Body weight was measured weekly for 10 weeks. For accurate food intake measurements and hence energy intake quantification, a custom-made, two-chamber Perspex insert was placed in each cage to ensure all food spillage could be collected and accounted for. Measurement of the amount of food consumed was performed once a week for 10 weeks and corrected for food spillage. Prior to collection estrous cycles and body composition were assessed. 10 weeks post-diet, 187 female mice were anesthetized using a 1:1 ratio of ketamine and xylazine, euthanized and blood and various tissues were collected and analyzed. All mouse experiments complied with all relevant ethical standards regarding animal research. All experiments were approved by the Sydney Local Health District Animal Welfare Committee within National Health and Medical Research Council guidelines for animal experimentation.

**Generation of PCOS mouse model**. PCOS was induced by peripubertal androgenization[25]. In short, 3-week-old C57Bl/6J female mice were implanted subcutaneously with a 1-cm SILASTIC brand implant (id, 1.47 mm; od, 1.95 mm, Dow Corning Corp, catalog no. 508–006) containing ~10 mg DHT or an empty pellet as a control. Silastic implants are made in-house and provide steady-state DHT release for at least 6 months[61]. At the time of animal collection all implants were removed and checked to ensure they still had DHT powder in them (which they did) and had not ruptured or leaked.

**Ovary preparation and morphological analysis**. Ovaries were dissected, weighed, fixed in 4% (weight/vol) paraformaldehyde overnight at 4 °C, and stored in 70% (vol/vol) ethanol before histological processing and analysis. Ovaries were processed through graded alcohols and embedded in glycol methacrylate resin (Technovit 7100; Heraeus Kulzer). 36 control and 35 PCOS ovaries (minimum 3 ovaries per treatment/diet) were assessed. Embedded ovaries were sectioned at 20 μm, stained with periodic acid-Schiff and counterstained with haematoxylin. For corpora lutea quantification, whole-section scans of every third section were taken under a light microscope using the DP70 Olympus camera.

**Assessment of estrous cycle**. Estrous-cycle stage was analyzed using vaginal epithelial cell smears taken daily for 11 consecutive days. Smears were collected using 15 μL of 0.9% sterile saline and transferred to glass slides to air dry. Dry smears were stained with 0.5% toluidine blue before examination under a light microscope. Estrous-cycle stage was determined based on the presence or absence of leukocytes, cornified epithelial cells, and nucleated epithelial cells. Proestrus was characterized by the presence of mainly nucleated and some cornified epithelial cells; at the estrous stage, mostly cornified epithelial cells were present; at metestrus, both cornified epithelial cells and leukocytes were present; and at diestrus, primarily leukocytes were present.

**DHT and testosterone levels**. Serum DHT and testosterone levels (extracts of 100 μl of mouse serum) were assessed by liquid chromatography-tandem mass spectrometry (LC-MS/MS)[62]. The limits of quantitation (defined as the lowest level that can be detected with a CV of <20%) were 0.05 ng/ml for DHT and 0.01 ng/ml for testosterone.

**LH and FSH levels**. Blood was collected from females by cardiac exsanguination under ketamine/xylazine anesthesia, and collected serum was stored at −20 °C. Mice were collected at the diestrus stage for cycling mice and pseudo-diestrus for acyclic PCOS mice. FSH and LH levels in 10 μl of sera diluted in 15 μl serum dilution buffer (1 in 2.5 dilution) was measured using EMD Millipore's MILLIPLEX® MAP 96-well Mouse Pituitary Magnetic Bead custom Kit (Cat# MPTMAG-49K) according to manufacturer's instructions (EMD Millipore, Billerica, MA, USA). Luminex MAGPIX instrument (Luminex Corporation, Northbrook, IL, USA) calibrated with MAGPIX Calibration and Performance Verification Kits (Millipore) with xPONENT software (Luminex) was used to acquire data. Saved data was then analyzed as the Median Fluorescent Intensity (MFI) using spline curve-fitting for calculating analyte concentrations in samples with Multiplex Analyst software Version 5.1 (Luminex).

**Body composition**. Body composition was assessed in all mice by dual-energy X-ray absorptiometry (DEXA) using the GE PIXImus2 Series Densitometer (GE

Medical Systems Ultrasound and BMD) under isoflurane inhalation anesthetic on week 10 of the diet.

**Adipose tissue analysis**. Parametrial fat pads were weighed, fixed in 4% paraformaldehyde, embedded in paraffin, sectioned at 8 μm and stained with haematoxylin and eosin. To assess adipocyte cell size, five different pictures were taken from each of three sections of the fat pad, with a minimum of 200 μm separating these sections. Images were taken at ×40 magnification under the Olympus BX60 light microscope for histomorphometric analysis. Adipocyte area was quantified using ImageJ version 1.51 software (NIH). All parametrial fat pads were analyzed without knowledge of treatment group.

**Adiponectin assay**. Total full-length mouse adiponectin were measured in serum using a Quantikine ELISA Kit from R&D Systems (MRP300) according to manufacturer's instructions. The mean minimum detectable dose of mouse adiponectin was 0.003 ng/ml.

**Cholesterol and triglyceride levels**. Serum total cholesterol and triglyceride levels were assayed enzymatically with kits obtained from Wako (Cholesterol E Kit, 439-17501; Triglyceride Kit, 432-40201).

**Fasting glucose levels and glucose tolerance test**. Fasting glucose levels and intraperitoneal glucose tolerance test (ipGTT) were measured on week 10 of the diet. Mice were fasted for 6 h before a baseline blood glucose reading, followed by an i.p. injection of glucose at 2 g/kg BW. Blood glucose was then measured at 15, 30, 60, and 90 min. Blood was obtained from a tail vein puncture, and blood glucose was measured using glucose strips on an Accu-Chek glucometer (Roche).

**Metabolic cage and diet preference study**. Four month old C57Bl/6J female mice (17 control and 16 DHT-induced PCOS) were housed individually in metabolic chambers (Mouse Promethion caging system, Sable Systems™, Las Vegas, USA) and maintained under standard housing conditions (ad libitum access to chow food and water in a temperature-controlled and humidity-controlled, 12-h light/dark environment) at the ANZAC Research Institute. 72 h (3 dark and 3 light cycles) of measurements were included for analysis and were taken after mice were acclimatized for 2 days. Each cage had flow-through respirometer to monitor metabolic rates, and sensor technology to monitor food, water uptake and body mass. Movement was also detected using infrared beam breaks in axis $X$, $Y$, and $Z$. Mice were then housed in standard cages and maintained under standard housing conditions for 1 week. Subsequently, these same mice were provided with ad libitum access to three different diets with either high protein (60% P, 20% C, 20% F), high carbohydrate (75% C, 5% P, 20% F) or high fat (75% F, 5% P, 20% C) content (Specialty Feeds) for 7 days (2 days acclimatization, 5 days of measurement). Food intake was measured daily for 4 days and body weight recorded at the start and end of exposure to the experimental diet. To account for food waste, a custom designed 2-chambre Perspex insert was installed beneath the food hopper of each cage which also allowed for the 3 diet separation within the cage for accurate measurement.

**Statistical modeling and analysis**. Macronutrient and food intake were analyzed based on average intake per mouse per cage per day. All other outcomes are modeled for individual animals. Data that were not normally distributed were log transformed before analysis or handled using appropriate link-functions. Nondetectable DHT steroid samples were treated as the value set for the limit of detection corrected with a data substitution method to minimize left censoring bias in serum steroid measurements[63]. Nondetectable LH and FSH samples were treated as the lowest value detected. Data are presented using the Geometric Framework approach, visualized through response surfaces and analyzed using generalized additive modeling (GAM) using the *mgcv* package in R v3.5.1., and through mixture-models using the *mixexp* package[64,65]. All surfaces are color coded across both control and PCOS groups, with red the highest elevation and blue the lowest in the pooled data from both surfaces. For data using counts, GAMs were fitted with the negative binomial distribution (log-link). In all cases, surfaces are presented on the raw scale to aid in interpretation (i.e., back transformed after model fitting). Comparisons between control and PCOS treatment groups were performed using Prism 8 software (GraphPad Software, Inc). Data that was normally distributed was analyzed using a Student's *t*-test, while a non-parametric Mann–Whitney test was used for data that was not normally distributed. Proportions were analyzed by Fisher's exact test. $P$ values <0.05 were considered statistically significant. Comparisons are presented as violin plots with median and quartile values indicated by dotted lines and a bar graph for proportional data. Data values in the result section are presented as mean ± SEM.

**Surface subtraction**. For each overlapping point on the two predicted surfaces we estimated the difference ($d$) as:

$$d = y_1 - y_2, \quad (1)$$

where $y_1$ is the estimate on the first surface and $y_2$ is the estimate on the second. To

estimate the significance of each point of difference we calculate a confidence limit (CFL) around the difference and, where that CFL excluded zero we inferred a significant difference. CFLs were calculated via Eqs. (2) and (3):

$$CFL = d \pm SE_d \times q, \quad (2)$$

$$SE_d = \sqrt{SE_1^2 + SE_2^2}, \quad (3)$$

where $q$ is a constant specifying the width of the CFL (1.96, 2.58, and 3.29, for 95, 99, and 99.9%, respectively) and $SE_1$ and $SE_2$ are the predicted standard errors for the points on the two surface[66,67]. In cases where outcomes were transformed, or modeled using a link function, CFLs were constricted on the latent scale, before overall surfaces were back transformed for interpretation.

**Reporting summary**. Further information on research design is available in the Nature Research Reporting Summary linked to this article.

## Data availability

The raw data that supports the findings in this study has been deposited in a publicly accessible database [https://github.com/AlistairMcNairSenior/GFN_PCOS]. Source data are provided with this paper.

## Code availability

Custom codes have been deposited in a publicly accessible database [https://github.com/AlistairMcNairSenior/GFN_PCOS].

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

## Acknowledgements

We thank Victoria Andriessen, Jenny Spaliviero, Mamdouh Khalil, and the staff of the ANZAC animal facility for experimental support. This study was support by Project Support Funding from the ANZAC Research Institute, and by support from the School of Women's and Children's Health, University of New South Wales Sydney, including for a PhD Scholarship awarded to Valentina Rodriguez Paris. Samantha M. Solon-Biet is supported by the NHMRC Peter Doherty Biomedical Fellowship (no. GNT1110098) and the University of Sydney SOAR fellowship.

## Author contributions

V.R.P., S.M.S.-B., S.J.S., D.J.H., K.A.W. designed research, V.R.P., S.M.S.-B., M.C.E., M.J.C., R.D., N.T. performed research, V.R.P., S.M. S.-B., A.M.S., K.A.W. analyzed data, V.R.P., S.M.S.-B., A.M.S., M.C.E., M.J.C., W.L.L., R.B.G., S.J.S., D.J.H., K.A.W. wrote the manuscript.

## Competing interests

The authors declare no competing interests.
