## [Peer Review File · Nature Communications]

REVIEWER COMMENTS

Reviewer #1 (Remarks to the Author):

Women with PCOS are more likely obese and lifestyle interventions including diet and exercise are first-line treatment. There is to date few studies investigating dietary impact on PCOS related symptoms and a recent review conclude that there is low grade of evidence to draw conclusion of optimal dietary composition in the management of PCOS.

The present manuscript used an approach that was previously published in Cell Metabolism and fed dihydrotestosterone (DHT) induced PCOS-like mice for 10 weeks. They found that there is an optimum macronutrient balance that can ameliorate ovulation, as shown by improved estrous cycles and corpus luteum. But, the metabolic outcomes in the PCOS-like mice were not improved by any diet combination.

Overall, the manuscript suggests a novel approach to define an optimal dietary combination and it is well-written and the methodology of good quality. As the prepubertal DHT induced PCOS model develop obesity, insulin resistance, cardiac hypertrophy, increase in liver triglycerides and lipid content among many metabolic features, and because the study investigate different diets that previously has been shown to have an impact on cardiometabolic function, it is surprising why the metabolic phenotyping is sparse.

Why not performing OGTT? Measure liver TG content? Perform echocardiography? Just to mention few suggested methods that are missing. Therefore, additional experiment aiming to improve the metabolic outcomes would be relevant, especially as authors discuss extensively about weight loss/hypocaloric diet.

Specific comments:

For each variable there one or two dietary composition were the most optimal, include the results in the plots, e.g. in Figure 1a, and perform statistics to show if its reversed or not. Furthermore, for each graph in all figures, include number of mice used.

Figure 1f: The representative figures/graphs must be complemented with figures including each estrus cycle stage, number of mice and whether there is statistically significance between controls and PCOS and if its reversed by diet including the most successful diet composition.

Only DHT was analyzed. However, DHT is administered to the mice and therefore it would be relevant to measure circulating testosterone and estradiol. If not possible, steroidogenic activity/enzymes/transcription factors can be analyzed in ovaries, adipose tissue and liver as an example.

Suppl. fig 1: Could authors elaborate more on the selection of these diet combinations? For example, why did you choose the extreme 5% and 60% of protein? If shown effective, could these diet combinations be followed by women in the long-term and is there any evidence that they are safe to follow?

Line 186-187: Mention the significantly lower locomotor activity in PCOS mice during daytime, even if this difference cannot sufficiently explain the increased body weight in the PCOS mice.

Figure 4b: Did you analyze the adipocyte size separately for the different diet groups? Were the results similar to what you see when you combine them? Furthermore, material and methods miss detailed description of how adipocyte size was measured. Number of mice, sections analyzed etc.

Line 265-267: I missed the numbering here. Does it mean that 15 mice had estrous cycles but 8 of them had CL? Need to be clear in the results and discussion.

Line 287-288: The benefits of the Mediterranean diet could be attributed to the food composition and the food choices, rather than the actual percentage of macronutrient content. It is important to highlight this difference.

Line 302-305: As discussed here, a hypocaloric diet reducing weight is shown to be more effective rather than macronutrient content in women with PCOS. It would be interesting to test whether weight loss in the PCOS mice combined or not with the optimal dietary matrix identified in the current study could lead to an increased ovulation ratio, and to changes in the metabolic function.

The metabolic complications are tightly associated to PCOS and as shown in the current study they cannot be rescued by adherence to a specific macronutrient matrix. The importance of weight loss and exercise should be highlighted at the end of the discussion, as it is of high clinical relevance.

Statistics

I am not familiar with the GAM and mgcv package in R. However, for phenotypic analyses, as suggested for each variable include the diet composition that is most effective in ameliorating PCOS-like phenotypic changes and perform a one-way ANOVA and tukey or dunnet's post hoc test.

Reviewer #3 (Remarks to the Author):

The authors Rodriguez Paris et al. describe a sophisticated nutritional approach -the Geometric Framework- and analyses focused on macronutrient balance on PCOS traits using a DHT-induced experimental PCOS mouse model. The data presented are technically sound, the results are novel, and it represents important scientific insights. Their conclusions are justified by the results obtained. Nevertheless, some minor aspects are potentially needing attention, in order to either clarify specific statements or provide an adequate detailed level of description. These are listed below:

1. Concluding sentence in abstract for instance is valid for the specific mouse model used (DHT-induced PCOS model), which is not necessarily translatable to all other PCOS conditions. Careful phrasing is suggested.
2. Can the authors provide also the lean mass of the control and PCOS mice, in order to show a clear view whether it is specifically only adipose tissue that is increased in PCOS mice? In fact, measurements of faecal excretion might be worthwhile to add to obtain an adequate energy

balance. Based on the data shown, the significant decreased activity in the light phase in PCOS mice might -at least partly- contribute to the observed increase in body weight.

3. Table 1: is energy expenditure (EE) corrected for total body weight or lean mass (latter is considered superior, although ANCOVA analysis is preferred in case lean mass is significantly different)? Alternatively, please add uncorrected EE.

3. Can the authors please provide more details on the procedure of "accurate food and energy intake" (line 409)? It seems that based on its description, either 4 or 5 cages were used per dietary group for food intake measures, so n=4-5. This is subsequently used for most of the figure panels with 10 dietary groups per Control or PCOS group, right? Would individual housing and exact measurements of also food intake not provide much more statistical power and biological relevance for these groups (n=100 versus n=40)? Indeed, mice in the indirect calorimetry system were housed individually, known to increase food intake. It is difficult to trace whether a similar food intake was seen in the group-housed mice versus individual housed mice. Can authors confirm or deny?

4. As these mice are very young at start and still growing, why did some of the diets not adhere to the AIN-93G (or even -93M) recommendations, especially for minimal protein content? Was this intended? Linked to this topic, can the authors please clarify whether dietary composition of individual nutrients of e.g. carbohydrate sources remained at an identical ratio when %C was altered (diet 1 vs 2 for example)? Same applies to other macronutrients.

5. Discussion, line 270-291: it appears that the majority of studies using dietary interventions are focused on weight loss and thereby improving also PCOS parameters in overweight/obese women; if this is the case, are the present findings applicable only to DHT-induced PCOS, since these female mice cannot be considered obese?

Finally, some very minor textual typo's:

* The usage of RQ is phrased more adequate as RER, since whole body measurements are presented.

* line 257: sentence lacks 'of' between development and PCOS.

* line 286: 39% F, lacks F

* is it HoMA-IR (line 291) or HOMA-IR? Likewise, oGTT (line 355) or OGTT?

* decreased, not deceases (line 389)

* line 430: ovaries were sectioned, delete word and

We are most grateful for the comments that our paper is well-written, technically sound and presents novel data with important scientific insights. We thank the reviewers for their most helpful and constructive suggestions which we have incorporated into the revision, such that all issues raised have been resolved and the manuscript improved. Our itemized and detailed responses to their comments follows. We would, therefore, be grateful if our revised manuscript could be further considered for publication in Nature Communications.

RESPONSES TO REFEREES

New data added:

Figure 1f-h and Supplementary Figure 4: We have carried out additional analysis to provide additional data on estrous cyclicity in control and PCOS mice.

Figure 2d: We have carried out additional analysis to provide lean mass in control and PCOS mice.

Figure 4c: We have carried out additional analysis to now provide adipocyte area response to diet in control and PCOS mice.

Supplementary Figure 5: We have undertaken an additional experiment to assess circulating triglyceride levels.

Supplementary Figure 6: We have carried out additional analysis of our data to assess GTT response.

Updated results section: We have carried out additional analysis of our data to assess circulating T levels in control and PCOS mice.

Note: *Reviewers comments verbatim in italics.* Rebuttal in regular font. Pages and line numbers are from the revised manuscript with highlighted corrections shown in the revised manuscript by red coloured text.

Manuscript ID# NCOMMS-20-07738-T

REVIEWER COMMENTS

Reviewer #1:

1. Women with PCOS are more likely obese and lifestyle interventions including diet and exercise are first-line treatment. There is to date few studies investigating dietary impact on PCOS related symptoms and a recent review conclude that there is low grade of evidence to draw conclusion of optimal dietary composition in the management of PCOS.

The present manuscript used an approach that was previously published in Cell Metabolism and fed dihydrotestosterone (DHT) induced PCOS-like mice for 10 weeks. They found that there is an optimum macronutrient balance that can ameliorate ovulation, as shown by improved estrous cycles and corpus luteum. But, the metabolic outcomes in the PCOS-like mice were not improved by any diet combination.

Overall, the manuscript suggests a novel approach to define an optimal dietary combination and it is well-written and the methodology of good quality. As the prepubertal DHT induced PCOS model develop obesity, insulin resistance, cardiac hypertrophy, increase in liver triglycerides and lipid content among many metabolic features, and because the study investigate different diets that previously has been shown to have an impact on

cardiometabolic function, it is surprising why the metabolic phenotyping is sparse.

Response: We thank the reviewer for their positive comments on our manuscript. Based on data from our previous Cell Metabolism paper where dietary macronutrient balance had a significant impact on metabolic outcomes, the hypothesis of this study was that manipulation of macronutrient balance would have a significant impact on PCOS-associated metabolic outcomes. However, what we identified was that hyperandrogenic PCOS mice exhibit a limited ability for their metabolic system to respond to variations in diet. In response to Reviewer 1's comment we have undertaken additional experiments to further assess the impact of diet on the metabolic phenotype of the PCOS mouse model. New data on the metabolic outcomes of lean mass (Fig. 2d), adipocyte size (Fig. 4c), circulating triglyceride levels (Supplementary Fig. 5) and glucose tolerance test (GTT) response (Supplementary Fig. 6) have now been added. We have revised the manuscript as follows:

Page 7, lines 164-166

The increased weight gain in PCOS females was reflected by an increase in body fat content ($P < 0.001$, ~~Fig. 2b, c and d~~) and also lean mass ($P < 0.001$) (Fig. 2b-e).

Page 21-22, lines 498-506

Adipose tissue analysis

Parametrial fat pads were weighed, fixed in 4% paraformaldehyde, embedded in paraffin, ~~and~~ sectioned at 8 μm ; ~~and Sections were~~ stained with haematoxylin and eosin. ~~To assess adipocyte cell size, five different pictures were taken from each of three sections of the fat pad, with a minimum of 200 μm separating these sections. and Images were taken at 40x magnification under the Olympus BX60 light microscope for histomorphometric analysis. Five different pictures were taken from each of three sections of the fat pad, with a minimum of 200 μm separating these sections.~~ Adipocyte area was quantified using ImageJ version 1.51 software (NIH). All parametrial fat pads were analyzed without knowledge of treatment group.

Page 9, lines 205-211

When data across all diets was combined, hyperandrogenism induced a significant increase in adipocyte area within PCOS parametrial fat pads compared to control mice ($1729 \pm 54.3 \mu\text{m}^2$ vs. $1573 \pm 50.6 \mu\text{m}^2$) ($P = 0.04$, Fig. 4a and b). This finding is in agreement with previous mouse PCOS studies (Benrick et al., 2017, Caldwell et al., 2017). ~~Analysis of changes in response to diet for adipocyte size, revealed the same pattern as changes in response to diet for body weight between control and PCOS mice. PCOS mice were more sensitive to low macronutrient intakes, but otherwise both groups displayed comparable patterns of response to macronutrient intake (Fig. 4c).~~

Page 22, lines 513-515

Cholesterol and triglyceride levels

Serum total cholesterol ~~and triglyceride~~ levels were assayed enzymatically with kits obtained from Wako (Cholesterol E Kit, 439-17501; ~~Triglyceride Kit, 432-40201~~).

Page 10-11, lines 238-244

~~Irrespective of diet and consistent with our previous study (Caldwell et al., 2017) PCOS mice did not display a significant increase in serum triglyceride levels (Supplementary Fig. 5a). Triglycerides levels in control and PCOS mice were driven by P intake ($P < 0.05$, Supplementary Fig. 5b), and P and C intakes ($P < 0.05$, Supplementary Fig. 5c), respectively,~~

but overall levels were within a similar range for both groups (40-65 mg/dL). In line with this, analysis of changes in response to diet between control and PCOS mice showed overall comparable responses to diet between both treatment groups (Supplementary Fig. 5d).

Page 22, lines 517-522

Fasting glucose levels and glucose tolerance test

Fasting glucose levels and intraperitoneal glucose tolerance test (ipGTT) were measured on week 10 of the diet. Mice were fasted for 6 h before a baseline blood glucose reading, followed by an i.p. injection of glucose at 2 g/kg BW. Blood glucose was then measured at 15, 30, 60 and 90 min. Blood was obtained from a tail vein puncture ~~piere~~, and blood glucose was measured using glucose strips on an Accu-Chek glucometer (Roche).

Page 11-12, lines 263-268

GTT response was also assessed and irrespective of diet and consistent with our previous studies (Caldwell et al., 2014, Caldwell et al., 2017), PCOS mice did not display an altered GTT response to control mice (Supplementary Fig. 6a), with both groups displaying similar GTT AUC across the majority of dietary intakes (Supplementary Fig. 6b and 6c). Correspondingly, analysis of changes in response to diet between control and PCOS mice identified no difference in response to diet between both groups for GTT response (Supplementary Fig. 6d).

2. Why not performing OGTT? Measure liver TG content? Perform echocardiography? Just to mention few suggested methods that are missing. Therefore, additional experiment aiming to improve the metabolic outcomes would be relevant, especially as authors discuss extensively about weight loss/hypocaloric diet.

Response: We have now added new metabolic outcome data on lean mass (Fig. 2d), adipocyte size (Fig, 4c), circulating triglyceride levels (Supplementary Fig. 5) and glucose tolerance test (GTT) response (Supplementary Fig. 6), please see Reviewer 1, point 1 for details. Echocardiography in mice requires a very skilled sonographer with a specialised ultrasound and probe, hence this outcome was beyond the scope of this study. The aim of this study was to identify the impact of macronutrient balance on PCOS features in a rigorously controlled environment setting. We agree that in the future, an additional experiment testing whether diets aimed at weight loss in the PCOS mice would be interesting. However, as the PCOS mouse model takes over 3 months to set up and an additional 3-6 months would be required for analysis of the endpoints, this experiment is beyond the scope of the current study.

3. For each variable there one or two dietary composition were the most optimal, include the results in the plots, e.g. in Figure 1a, and perform statistics to show if its reversed or not. Furthermore, for each graph in all figures, include number of mice used.

Response: In the current study we identified an optimum dietary macronutrient balance of a low protein, medium carbohydrate and fat diet can ameliorate the key PCOS reproductive trait of anovulation. We initially assessed our data grouped by control and PCOS, and then used 3D generalized additive model (GAM) response surfaces to investigate the relationship between macronutrient intake and PCOS-like traits observed. The regions of the optimal dietary composition are encircled in red in Figures 1c(iii) and d(iii). In contrast to PCOS reproductive traits, we identified that PCOS mice displayed a hindered ability for their metabolic system to respond to variations in diet, and varying macronutrient balance did not

have a significant beneficial effect on the development of metabolic PCOS traits. Therefore, no optimal dietary composition could be identified for each of the metabolic outcome, hence we have not encircled any areas in red. The statistics we have undertaken to show in each figure whether the PCOS trait has been reversed or not are shown by the difference in response between the control and PCOS mice (shown in Fig 1e(iii), Fig 1h, Fig 2i(iii), Fig 4c and 4g(iii), Fig 5d(iii), Fig 6d(iii)). In Fig 1e(iii) the area in grey shows where there is no statistically significant difference between control and PCOS mice and this correlates with the region of optimal dietary composition encircled in red (Fig 1d(iii)), hence there has, arguably, been a rescue of the PCOS trait as the two groups are not statistically different. Unlike in Fig 4g(iii) where the area is mostly pink/purple, showing the two groups are statistically different to each other across all diets, and hence diet has no beneficial effect in rescuing the PCOS trait. To make this clearer, we have added in additional text to Figure 1 legend where we have explained how to read all graphs. Figure legends have been revised to ensure number of mice used is included for each graph.

Page 31, lines 825-832

e, To determine differences in the presence of PCOS-like traits in response to diet, 3D GAM response surfaces are provided showing the difference in response between control and PCOS mice to macronutrients. 3D GAM response surfaces are provided showing the impact of main drivers (carbohydrate and fat) on CL numbers in control mice (**e(i)**), PCOS mice (**e(ii)**) and the response difference (**e(iii)**), demonstrating a comparable ovulatory response between control and PCOS above a C intake threshold of >20 kJ/day and F threshold of >25 kJ/day ~~F~~. Areas in grey indicate no significant response difference, while pink/purple areas indicate a significant response difference between PCOS and control mice.

4. Figure 1f: The representative figures/graphs must be complemented with figures including each estrus cycle stage, number of mice and whether there is statistical significance between controls and PCOS and if its reversed by diet including the most successful diet composition.

Response: As requested, we have now added new data showing the percentage of control and PCOS mice cycling (Supplementary Figure 4a), the percentage of time control and PCOS mice spent at each stage of the estrous cycle (Figure 1f) and if diet could reverse the PCOS feature of acyclicity (Figure 1h). Data shows that overall, there is a significant difference between PCOS and control mice, with fewer PCOS mice cycling (Supplementary Figure 4a). However, a proportion of the PCOS mice did cycle, demonstrated clearly by the finding that some PCOS mice were observed to spend time at all stages of the estrous cycle (Figure 1f and Supplementary Figure 4a). Furthermore, analysis of changes in response to diet between control and PCOS mice, show diet can reverse the PCOS features of acyclicity. PCOS mice were observed to have comparable numbers of estrous cycles to control mice within a similar nutritional space as that observed for ovulatory response (Figure 1h and Supplementary Figure 4).

Page 7, lines 151-158

However, diet-induced restoration in ovulatory function in PCOS females was confirmed by examining estrous cycle patterns in control and PCOS mice (Fig. 1f-h ~~i-iii~~ and Supplementary Fig. 4). A proportion of PCOS mice cycled (15/94), demonstrated by the finding that some PCOS mice were observed to spend time at all stages of the estrous cycle (Fig. 1f). While the majority of PCOS mice were acyclic, ~~E~~corresponding with the macronutrient balance able to restore ovulatory function in PCOS mice, a subset of PCOS mice (~~15/94~~) within the intake

ranges of 20-30 kJ/day C and 15-25 kJ/day F displayed completed estrous cycles (Fig. 1g(iii) and Supplementary Fig. 4c(iii) fiii); ~~while all other PCOS mice were acyclic (Fig. 1fii).~~

Page 31-32, lines 832-841

f, % of time spent at each stage of the estrous cycle for control and PCOS mice, showing that although the majority of PCOS mice spent most time at the diestrus stage of the cycle, a subset of PCOS mice did cycle through proestrus, estrus and metestrus stages of the estrous cycle. **g**, Representative graphs of estrous-cycle patterns in control and PCOS mice, showing restoration of cyclicity in a PCOS mouse with a dietary intake of 20 kJ/day C and 21 kJ/day F, which falls within the range observed for PCOS mice exhibiting restored ovulations (**d(iii)**). **h**, Response difference between control and PCOS mice for the number of estrous cycles completed, demonstrating a comparable estrous cycle response between control and PCOS mice at similar C and F intakes that restore ovulatory response. **a, c-e**, n = 35-36 ovaries/treatment group; **f, h**, n = 93-94 mice/treatment group.

5. Only DHT was analyzed. However, DHT is administered to the mice and therefore it would be relevant to measure circulating testosterone and estradiol. If not possible, steroidogenic activity/enzymes/transcription factors can be analyzed in ovaries, adipose tissue and liver as an example.

Response: Serum steroid levels were measured using our liquid chromatography-tandem mass spectrometry (LC-MS/MS) method, which has a detection limit of 3-5 pg/ml. We have previously shown using this method that serum E2 levels in mice are undetectable at the diestrus stage (McNamara et al., 2010), hence E2 levels were unable to be reported for this study. However, we have now added new data assessing circulating testosterone levels. In agreement with our previous studies (Caldwell et al., 2014, Caldwell et al., 2017) serum testosterone levels were comparable between control (0.07 ± 0.01 ng/ml) and PCOS (0.05 ± 0.02 ng/ml) females.

Page 20, lines 475-479

DHT and testosterone levels

Serum DHT and testosterone levels (extracts of 100 μ l of mouse serum) were assessed by liquid chromatography-tandem mass spectrometry (LC-MS/MS), as previously described (Desai et al., 2019). The limits of quantitation (defined as the lowest level that can be detected with a CV of < 20%) was 0.05ng/ml for DHT and 0.01ng/ml for testosterone.

Page 5, lines 105-109

DHT treatment was validated by liquid chromatography-tandem mass spectrometry (LC-MS/MS), with PCOS mice (1.34 ± 0.17 ng/ml) exhibiting a significant increase in circulating DHT levels compared to control females (0.27 ± 0.04 ng/ml, $P < 0.001$). Serum testosterone levels were also assessed and in agreement with our previous studies (Caldwell et al., 2014, Caldwell et al., 2017), levels were observed to be comparable between control (0.07 ± 0.01 ng/ml) and PCOS (0.05 ± 0.02 ng/ml) females.

6. Suppl. fig 1: Could authors elaborate more on the selection of these diet combinations? For example, why did you choose the extreme 5% and 60% of protein? If shown effective, could these diet combinations be followed by women in the long-term and is there any evidence that they are safe to follow?

Response: We selected a wide range of diet combinations to enable us to discover the independent and interactive effects of macronutrients on PCOS traits. Diet combinations varying systematically in protein, carbohydrate and fat were carefully selected based on our previous publications (Solon-Biet et al., 2014, Solon-Biet et al., 2015) to allow optimal fitting of response surfaces to ensure we could generate a detailed nutrient landscape. This allowed us to identify a macronutrient combination that could rescue ovulatory dysfunction in PCOS mice. It is important to note that the extreme diets were not included as diet combinations that would be expected to be effective in ameliorating features of PCOS. Rather, the ranges of macronutrient values used in the selected diet combinations were chosen, based on our previous publications (Solon-Biet et al., 2014, Solon-Biet et al., 2015), for optimal power in fitting response surface models used for statistical analysis. Of note, in humans it is unlikely that long-term very low/very high protein intakes would be safe to follow. The Acceptable Macronutrient Distribution Range (AMDR) is 10-35% calories from protein and while a moderate reduction in protein intake has numerous health and ageing benefits, going beyond these proportions are not recommended. We have added additional text to the materials and methods section to clarify the rationale behind the selection of the diet combinations used.

Page 18-19, lines 429-434

At 7 weeks of age, control and DHT-induced PCOS mice were switched from standard chow diet and provided *ad libitum* access to one of 10 experimental diets varying in protein, carbohydrate, and fat content (Specialty Feeds). **The selection of macronutrient content used in the experimental diets was chosen, based on our previous publications (Solon-Biet et al., 2014, Solon-Biet et al., 2015), for optimal power in fitting response surface models used for statistical analysis (Supplementary Fig. 1 and Supplementary Table 1).**

7. Line 186-187: Mention the significantly lower locomotor activity in PCOS mice during daytime, even if this difference cannot sufficiently explain the increased body weight in the PCOS mice.

Response: As requested we have now added in a sentence mentioning the finding that PCOS mice exhibited lower day locomotor activity, but this decrease is insufficient to explain fully the increase in body weight observed in PCOS mice.

Page 9, lines 194-196

Although PCOS mice exhibited lower day locomotor activity, this difference was relatively small and unlikely to be sufficient to fully explain the increase in body weight in PCOS mice.

8. Figure 4b: Did you analyze the adipocyte size separately for the different diet groups? Were the results similar to what you see when you combine them? Furthermore, material and methods miss detailed description of how adipocyte size was measured. Number of mice, sections analyzed etc.

Response: Yes, we analyzed changes between the control and PCOS mice in response to the different diets for adipocyte size. Data revealed the same pattern as changes in response to diet for body weight between the two groups, with PCOS mice observed to be more sensitive to low macronutrient intakes. We apologise for not providing more detailed information on the methods used. This has now been added to the results and methods sections.

Page 9, lines 207-211

Analysis of changes in response to diet for adipocyte size, revealed the same pattern as changes in response to diet for body weight between control and PCOS mice. PCOS mice were more sensitive to low macronutrient intakes, but otherwise both groups displayed comparable patterns of response to macronutrient intake (Fig. 4c).

Page 21-22, lines 498-506

Adipose tissue analysis

Parametrial fat pads were weighed, fixed in 4% paraformaldehyde, embedded in paraffin, and sectioned at 8 μm . Sections were and stained with haematoxylin and eosin. To assess adipocyte cell size, five different pictures were taken from each of three sections of the fat pad, with a minimum of 200 μm separating these sections. and Images were taken at 40x magnification under the Olympus BX60 light microscope for histomorphometric analysis. Five different pictures were taken from each of three sections of the fat pad, with a minimum of 200 μm separating these sections. Adipocyte area was quantified using ImageJ version 1.51 software (NIH). All parametrial fat pads were analyzed without knowledge of treatment group.

9. Line 265-267: I missed the numbering here. Does it mean that 15 mice had estrous cycles but 8 of them had CL? Need to be clear in the results and discussion.

Response: Yes the reviewer has correctly understood this, 15 mice exhibited estrous cycles and 8 ovaries assessed displayed CL. It should be noted that all mice in the study were assessed for estrous cyclicity, but for assessment of CL populations (as is the case in all of our similar previous publications (Caldwell et al., 2017, Solon-Biet et al., 2015, Caldwell et al., 2014) and is common practice in the field) not all ovaries from every mouse were assessed. This is why in the manuscript we have provided the raw numbers as a total of the number actually assessed. Hence, in the results and discussion sections we have stated 8/35 ovaries displayed CL and 15/94 females exhibited complete estrous cycles.

10. Line 287-288: The benefits of the Mediterranean diet could be attributed to the food composition and the food choices, rather than the actual percentage of macronutrient content. It is important to highlight this difference.

Response: We agree this is an important point. We have modified the sentence to highlight this difference.

Page 13-14, lines 309-316

Notably, the dietary macronutrient ratio identified (14% P, 47% C and 39% F) falls within the ranges of a traditional Mediterranean diet, which has been associated with numerous health benefits such as improvement of lipid profile, protection against oxidative stress (Tosti et al., 2017, Fitó et al., 2007), decreased adiposity (Boghossian et al., 2013) and decreased risk of type-2 diabetes (Kolooverou et al., 2014), thus warranting further studies in the area of PCOS. Furthermore, a recent study has reported that women with PCOS with a high adherence to a Mediterranean diet presented with lower testosterone and HOMA-IR levels (Barrea et al., 2019). Thus, further studies defining if this beneficial effect on PCOS traits is attributed to the macronutrient balance alone or the food choices associated with a Mediterranean diet are warranted.

11. Line 302-305: As discussed here, a hypocaloric diet reducing weight is shown to be more effective rather than macronutrient content in women with PCOS. It would be interesting to

test whether weight loss in the PCOS mice combined or not with the optimal dietary matrix identified in the current study could lead to an increased ovulation ratio, and to changes in the metabolic function.

Response: We agree that it would be interesting to test whether weight loss in the PCOS mice combined or not with the optimal dietary matrix leads to an increased ovulation ratio and/or changes in metabolic function. However, as the PCOS mouse model takes over 3 months to set and up and an additional 3-6 months would be required for analysis of the endpoints this experiment is beyond the scope of this current study. Please also see Reviewer 1, point 2.

12. The metabolic complications are tightly associated to PCOS and as shown in the current study they cannot be rescued by adherence to a specific macronutrient matrix. The importance of weight loss and exercise should be highlighted at the end of the discussion, as it is of high clinical relevance.

Response: We have added in a new sentence to the discussion to highlight the importance of weight loss and exercise in the management of PCOS.

Page 18, lines 414-420

The finding that PCOS mice gained more weight at lower macronutrient intakes than control mice, supports implementation of the current clinical recommendations of weight loss and exercise for all women with PCOS. Hyperandrogenic PCOS mice **also** displayed a decreased capacity for their metabolic system to respond to variations in diet, and changes in macronutrient balance had a minimal beneficial effect on ameliorating metabolic dysfunction observed in PCOS mice. These findings support the notion that the underlying pathophysiology of PCOS has a predominant influence on the development of PCOS traits.

13. Statistics. I am not familiar with the GAM and mgcv package in R. However, for phenotypic analyses, as suggested for each variable include the diet composition that is most effective in ameliorating PCOS-like phenotypic changes and perform a one-way ANOVA and tukey or dunnet's post hoc test.

Response: The study was designed as a compositional analysis of the effects of macronutrients in three-dimensions (protein, carbohydrate and fat). This means the data should be analysed in terms of their response to a quantitative predictor (e.g. via a linear regression - of which a GAM is an advanced form). As this study was designed to estimate a quantitative trend over the nutrient space (i.e. with the different diets 'linking up to hold hands' across the space) it is inappropriate to make pair-wise comparisons between the responses on individual diets. The key problem in doing so, which is of particular relevance here is in terms of power; the sample size of any one diet was not designed to be large enough to detect differences between that diet and another. It is only through the entire array of diets that the study has the collective power to reliably detect effects. Simply showing a non-significant difference between PCOS and control animals on corresponding diets would not be convincing evidence that the two groups are 'the same' as the two groups would have small sample sizes. Please also see response to Reviewer 1, point 3 which is related to this point.

Reviewer #3:

1. The authors Rodriguez Paris et al. describe a sophisticated nutritional approach -the

Geometric Framework- and analyses focused on macronutrient balance on PCOS traits using a DHT-induced experimental PCOS mouse model. The data presented are technically sound, the results are novel, and it represents important scientific insights. Their conclusions are justified by the results obtained. Nevertheless, some minor aspects are potentially needing attention, in order to either clarify specific statements or provide an adequate detailed level of description. These are listed below:

Response: We thank the reviewer for their positive comments on our manuscript.

2. Concluding sentence in abstract for instance is valid for the specific mouse model used (DHT-induced PCOS model), which is not necessarily translatable to all other PCOS conditions. Careful phrasing is suggested.

Response: The concluding sentences in the abstract have been revised.

Page 2, lines 33-37

These findings ~~provide evidence~~ demonstrate that PCOS traits in a hyperandrogenic PCOS mouse model can be ameliorated selectively by diet, with reproductive PCOS traits displaying greater sensitivity than metabolic traits to dietary macronutrient balance. Hence, providing evidence to support the development of evidence-based dietary interventions is a promising strategy for the treatment of PCOS, especially reproductive traits.

3. Can the authors provide also the lean mass of the control and PCOS mice, in order to show a clear view whether it is specifically only adipose tissue that is increased in PCOS mice? In fact, measurements of faecal excretion might be worthwhile to add to obtain an adequate energy balance. Based on the data shown, the significant decreased activity in the light phase in PCOS mice might -at least partly- contribute to the observed increase in body weight.

Response: As requested we have carried out additional analysis and now provided additional data on lean mass in control and PCOS mice in Figure 2d. We did not collect any measurement of faecal excretion from this study. We have added in a sentence mentioning the finding that PCOS mice exhibited lower day locomotor activity - please also see response to Reviewer 1, point 7.

Page 7, lines 162-166

Regardless of dietary intervention, PCOS mice displayed a significant increase in body weight (23.5 ± 0.3 g) compared to control mice (21.7 ± 0.2 g) ($P < 0.001$, Fig. 2a), which is in agreement with previous studies using our hyperandrogenized PCOS mouse model (Caldwell et al., 2014, Caldwell et al., 2017). The increased weight gain in PCOS females was reflected by an increase in body fat content ($P < 0.001$, ~~Fig. 2b, c and d~~) and also lean mass ($P < 0.001$) (Fig. 2b-e).

Page 32, lines 847-848

d, Lean body mass (g) calculated by DEXA, showing that PCOS mice exhibit a significant increase in lean body mass compared to controls ($P < 0.001$).

4. Table 1: is energy expenditure (EE) corrected for total body weight or lean mass (latter is considered superior, although ANCOVA analysis is preferred in case lean mass is

significantly different)? Alternatively, please add uncorrected EE.

Response: Energy expenditure was corrected for total body weight, which is shown by the units of measure for energy expenditure in table 1 (kcal/12hr/kg). We have now specified this in the results section and the figure legend for table 1. Additionally, a correction has been made to the table unit description for EE as it was missing the value 12 before hr. As suggested, we have added data showing the uncorrected EE to Table 1, and statistical analysis showed there was no significant difference for uncorrected day and night EE expenditure between control and PCOS mice.

Page 8-9, Lines 189-194

Furthermore, metabolic cage analysis of the control and PCOS mice, 1 week prior to the dietary choice study, consuming a standard chow diet revealed there was no significant difference between the two groups for day and night food intake, indirect calorimetry measurements of energy expenditure, **body weight corrected energy expenditure**, oxygen consumption (vO_2), carbon dioxide expulsion (vCO_2), respiratory **exchange ratio (RER) quotient (RQ)**, and night locomotor activity (Table 1).

Page 35, Lines 915-920

Table 1 Metabolic state does not differ between control and PCOS mice. Measurements of indirect calorimetry by metabolic cages of control and PCOS mice on a standard chow diet, showing food intake (g), locomotor activity (beam breaks), **energy expenditure (kcal/12hr)**, **body weight corrected energy expenditure (kcal/12hr/kg)**, oxygen (O_2) consumption (VO_2), carbon dioxide (CO_2) produced (VCO_2), and respiratory **exchange ratio (RER) quotient (RQ)** do not differ between control and PCOS mice. Data are the mean \pm SEM; n = 16-17 mice/treatment group.

5. Can the authors please provide more details on the procedure of "accurate food and energy intake" (line 409)? It seems that based on its description, either 4 or 5 cages were used per dietary group for food intake measures, so n=4-5. This is subsequently used for most of the figure panels with 10 dietary groups per Control or PCOS group, right? Would individual housing and exact measurements of also food intake not provide much more statistical power and biological relevance for these groups (n=100 versus n=40)? Indeed, mice in the indirect calorimetry system were housed individually, known to increase food intake. It is difficult to trace whether a similar food intake was seen in the group-housed mice versus individual housed mice. Can authors confirm or deny?

Response: We apologise for not providing sufficient detail on the procedure used in this study. To ensure that accurate food intake measurements were obtained to allow for accurate energy intake quantification, a custom-made, two-chamber Perspex insert was placed in each cage to make sure all food spillage could be collected and accounted for. Measurement of the amount of food consumed was performed once a week for 10 weeks and corrected for food spillage. The reviewer is correct for each of the 10 dietary groups there were 4-5 cages of mice. For analysis, we initially compared between cages within each dietary group to ensure there were no large variations in food intake, before averaging intake per mouse per cage per day. We agree that potentially individual housing may allow more exact measurement of food intake, however, our animal ethics board would not approve female mice to be housed individually for such a long length of time as they are naturally social animals and are best kept in groups. We obtained approval to carry out individual housing of mice for the 5 days required for the separate metabolic cage study, as these cages are only designed to house 1

mouse at a time to allow accurate metabolic measurements. Mice individually housed in the metabolic cage study (standard chow diet) or the food choice study (3 diets available at once) were not on the same diet as the mice in the 10 experimental diet study. Therefore, it would not be appropriate to compare the food intake of mice between the different studies.

Page 18, lines 427-429

Mice were housed in groups of 2 or 3 mice per cage (4-5 cages per diet) and maintained under standard housing conditions (*ad libitum* access to food and water in a temperature- and humidity-controlled, 12-h light/dark environment) at the ANZAC Research Institute.

Page 19, lines 438-442

For accurate food intake measurements and hence energy intake quantification, a custom-made, two-chamber Perspex insert was placed in each cage to ensure all food spillage could be collected and accounted for food spillage. Food measurements of the amount of food consumed were performed once a week for 10 weeks and corrected for food spillage.

6. As these mice are very young at start and still growing, why did some of the diets not adhere to the AIN-93G (or even -93M) recommendations, especially for minimal protein content? Was this intended? Linked to this topic, can the authors please clarify whether dietary composition of individual nutrients of e.g. carbohydrate sources remained at an identical ratio when %C was altered (diet 1 vs 2 for example)? Same applies to other macronutrients.

Response: Experimental diets were started at 7 weeks of age to ensure mice were at an adult stage of their life. Experimental diets used are based on dietary formulations from our previous research (Solon-Biet et al., 2014, Solon-Biet et al., 2015). Diet compositions, which includes the extremes, were designed to ensure a wide range of diets spanning the viable dietary protein, carbohydrate and fat range in mice were tested. Limiting macronutrients to 21% protein content, for example, would only show a thin slice of the surface. By testing macronutrient ratios within a large nutrient space (shown in Supp fig 1), as we have done here, we were then able to construct response surfaces to determine how a wide range of macronutrient ratios influenced PCOS traits. The ratio of individual nutrient sources did not remain identical. However, starch remained the dominant carbohydrate across all diets. For protein, casein accounted for the majority of the diet, with minimal addition of methionine. Only one source of fat (soy oil) was used in all diets. Please also see response to Reviewer 1, point 6.

7. Discussion, line 270-291: it appears that the majority of studies using dietary interventions are focused on weight loss and thereby improving also PCOS parameters in overweight/obese women; if this is the case, are the present findings applicable only to DHT-induced PCOS, since these female mice cannot be considered obese?

Response: While the PCOS mice at the time of collection in this study were not obese, from previous studies using this mouse model, we know that DHT exposed females continually put on excess body weight and body weight does not plateau by the time of collection, 3 months after the start of DHT exposure (see Figure 1A in our previous publication showing body weight over weeks after DHT exposure (Bertoldo et al., 2019)). Therefore, the mice in the current study would have likely become obese with time and the mechanisms involved would already have been perturbed. Numerous studies have demonstrated that weight loss in women suffering from PCOS improves clinical outcomes (Kataoka et al., 2017, Balen et al., 2016).

However, despite recommendations that diet and lifestyle are front-line therapy for women with PCOS, unfortunately, at present there is a paucity of well-controlled clinical studies with high enough and well sustained compliance on the impact of dietary composition on PCOS to answer the question in humans at this time. Due to the scarcity of high-quality clinical research in this area, we used a well characterised hyperandrogenic PCOS mouse model, that displays a breadth of reproductive and metabolic PCOS traits, as a good model, and starting point to develop testable hypotheses for humans. The findings from this current study provide a solid evidence-based basis for the development of testable nutritional strategies to be investigated in other PCOS animal models and translated into human clinical studies.

8. *Finally, some very minor textual typo's:*

**The usage of RQ is phrased more adequate as RER, since whole body measurements are presented.*

Response: Corrected as requested, see Page 9, 15 and 35, lines 193, 343, 919 and Table 1

9. ** line 257: sentence lacks 'of' between development and PCOS.*

Response: Corrected as requested, see Page 12, lines 280

10. ** line 286: 39% F, lacks F*

Response: Corrected as requested, see Page 13, lines 309

11. ** is it HoMA-IR (line 291) or HOMA-IR? Likewise, oGTT (line 355) or OGTT?*

Response: Corrected as requested, see Page 14 and 16, lines 314 and 380

12. ** decreased, not deceases (line 389)*

Response: Corrected as requested, see Page 18, lines 417

** line 430: ovaries were sectioned, delete word and*

Response: Corrected as requested, see Page 20, lines 461

Thank you very much for your time and effort in considering this manuscript.

Yours sincerely,

Associate Professor Kirsty Walters

References used in rebuttal

- BALEN, A. H., MORLEY, L. C., MISSO, M., FRANKS, S., LEGRO, R. S., WIJAYARATNE, C. N., STENER-VICTORIN, E., FAUSER, B. C., NORMAN, R. J. & TEEDE, H. 2016. The management of anovulatory infertility in women with polycystic ovary syndrome: an analysis of the evidence to support the development of global WHO guidance. *Hum Reprod Update*, 22, 687-708.
- BARREA, L., ARNONE, A., ANNUNZIATA, G., MUSCOGIURI, G., LAUDISIO, D., SALZANO, C., PUGLIESE, G., COLAO, A. & SAVASTANO, S. 2019. Adherence to the Mediterranean Diet, Dietary Patterns and Body Composition in Women with Polycystic Ovary Syndrome (PCOS). *Nutrients*, 11.
- BENRICK, A., CHANCLON, B., MICALLEF, P., WU, Y., HADI, L., SHELTON, J. M., STENER-VICTORIN, E. & WERNSTEDT ASTERHOLM, I. 2017. Adiponectin protects against development of metabolic disturbances in a PCOS mouse model. *Proceedings of the National Academy of Sciences*, 114, E7187-e7196.
- BERTOLDO, M. J., CALDWELL, A. S. L., RIEPSAMEN, A. H., LIN, D., GONZALEZ, M. B., ROBKER, R. L., LEDGER, W. L., GILCHRIST, R. B., HANDELSMAN, D. J. & WALTERS, K. A. 2019. A Hyperandrogenic Environment Causes Intrinsic Defects That Are Detrimental to Follicular Dynamics in a PCOS Mouse Model. *Endocrinology*, 160, 699-715.
- BOGHOSSIAN, N. S., YEUNG, E. H., MUMFORD, S. L., ZHANG, C., GASKINS, A. J., WACTAWSKI-WENDE, J., SCHISTERMAN, E. F. & BIOCYCLE STUDY, G. 2013. Adherence to the Mediterranean diet and body fat distribution in reproductive aged women. *European journal of clinical nutrition*, 67, 289-294.
- CALDWELL, A. S., MIDDLETON, L. J., JIMENEZ, M., DESAI, R., MCMAHON, A. C., ALLAN, C. M., HANDELSMAN, D. J. & WALTERS, K. A. 2014. Characterization of reproductive, metabolic, and endocrine features of polycystic ovary syndrome in female hyperandrogenic mouse models. *Endocrinology*, 155, 3146-59.
- CALDWELL, A. S. L., EDWARDS, M. C., DESAI, R., JIMENEZ, M., GILCHRIST, R. B., HANDELSMAN, D. J. & WALTERS, K. A. 2017. Neuroendocrine androgen action is a key extraovarian mediator in the development of polycystic ovary syndrome. *Proceedings of the National Academy of Sciences*, 114, E3334.
- DESAI, R., HARWOOD, T. & HANDELSMAN, D. 2019. Simultaneous Measurement of 18 Steroids in Human and Mouse Serum by Liquid Chromatography–Mass Spectrometry without Derivatization to Profile the Classical and Alternate Pathways of Androgen Synthesis and Metabolism. *Clinical Mass Spectrometry*, 11.
- FITÓ, M., GUXENS, M., CORELLA, D., SÁEZ, G., ESTRUCH, R., DE LA TORRE, R., FRANCÉS, F., CABEZAS, C., LÓPEZ-SABATER, M. D. C., MARRUGAT, J., GARCÍA-ARELLANO, A., ARÓS, F., RUIZ-GUTIERREZ, V., ROS, E., SALAS-SALVADÓ, J., FIOL, M., SOLÁ, R., COVAS, M.-I. & FOR THE, P. S. I. 2007. Effect of a Traditional Mediterranean Diet on Lipoprotein Oxidation: A Randomized Controlled Trial. *Archives of Internal Medicine*, 167, 1195-1203.
- KATAOKA, J., TASSONE, E. C., MISSO, M., JOHAM, A. E., STENER-VICTORIN, E., TEEDE, H. & MORAN, L. J. 2017. Weight Management Interventions in Women with and without PCOS: A Systematic Review. *Nutrients*, 9, 996.
- KOLOVEROU, E., ESPOSITO, K., GIUGLIANO, D. & PANAGIOTAKOS, D. 2014. The effect of Mediterranean diet on the development of type 2 diabetes mellitus: A meta-analysis of 10 prospective studies and 136,846 participants. *Metabolism*, 63, 903-911.
- MCNAMARA, K. M., HARWOOD, D. T., SIMANAINEN, U., WALTERS, K. A., JIMENEZ, M. & HANDELSMAN, D. J. 2010. Measurement of sex steroids in murine blood and reproductive tissues by liquid chromatography-tandem mass spectrometry. *J. Steroid Biochem. Mol. Biol*, 121, 611-618.
- SOLON-BIET, S. M., MCMAHON, A. C., BALLARD, J. W., RUOHONEN, K., WU, L. E., COGGER, V. C., WARREN, A., HUANG, X., PICHAUD, N., MELVIN, R. G., GOKARN, R., KHALIL, M., TURNER, N., COONEY, G. J., SINCLAIR, D. A., RAUBENHEIMER, D., LE COUTEUR, D. G. & SIMPSON, S. J.

2014. The ratio of macronutrients, not caloric intake, dictates cardiometabolic health, aging, and longevity in ad libitum-fed mice. *Cell Metab*, 19, 418-430.
- SOLON-BIET, S. M., WALTERS, K. A., SIMANAINEN, U. K., MCMAHON, A. C., RUOHONEN, K., BALLARD, J. W., RAUBENHEIMER, D., HANDELSMAN, D. J., LE COUTEUR, D. G. & SIMPSON, S. J. 2015. Macronutrient balance, reproductive function, and lifespan in aging mice. *Proc. Natl. Acad. Sci. U. S. A.*, 112, 3481-3486.
- TOSTI, V., BERTOZZI, B. & FONTANA, L. 2017. Health Benefits of the Mediterranean Diet: Metabolic and Molecular Mechanisms. *The Journals of Gerontology: Series A*, 73, 318-326.

REVIEWERS' COMMENTS:

Reviewer #1 (Remarks to the Author):

Authors have adressed all comments and concerns and I have nothing more to add.

Reviewer #3 (Remarks to the Author):

The authors adequately incorporated adaptations which resolved all raised issues and improved the manuscript scientifically and for its readability and clarification for readers of Nature Communications. I have no additional comments or suggestions.

REVIEWER COMMENTS

Reviewer #1:

1. Authors have adressed all comments and concerns and I have nothing more to add.

Response: We thank the reviewer for their time, help and constructive recommendations to improve our manuscript. We are glad the reviewer considers their concerns have been answered and that all comments have been addressed appropriately.

Reviewer #3:

1. The authors adequately incorporated adaptations which resolved all raised issues and improved the manuscript scientifically and for its readability and clarification for readers of Nature Communications. I have no additional comments or suggestions.

Response: Once again, we thank the reviewer for their time and constructive suggestions to improve our manuscript. We are pleased the reviewer believes we have adequately incorporated adaptations to resolve the issues raised in order to improve the manuscript.

Thank you very much for your time in considering this manuscript.

Yours sincerely,

Associate Professor Kirsty Walters